# Efficient learning of nonlinear prediction models with time-series privileged information

**Bastian Jung**
Chalmers University of Technology
mail@bastianjung.com

**Fredrik D. Johansson**
Chalmers University of Technology
fredrik.johansson@chalmers.se

## Abstract

In domains where sample sizes are limited, efficient learning algorithms are critical. Learning using privileged information (LuPI) offers increased sample efficiency by allowing prediction models access to auxiliary information at training time which is unavailable when the models are used. In recent work, it was shown that for prediction in linear-Gaussian dynamical systems, a LuPI learner with access to intermediate time series data is never worse and often better in expectation than any unbiased classical learner. We provide new insights into this analysis and generalize it to nonlinear prediction tasks in latent dynamical systems, extending theoretical guarantees to the case where the map connecting latent variables and observations is known up to a linear transform. In addition, we propose algorithms based on random features and representation learning for the case when this map is unknown. A suite of empirical results confirm theoretical findings and show the potential of using privileged time-series information in nonlinear prediction.

## 1 Introduction

In data-poor domains, making the best use of all available information is central to efficient and effective machine learning. Despite this, supervised learning is often applied in such a way that informative data is ignored. A good example of this is learning to predict the condition of a patient at a set follow-up time based on information of the first medical examination. Classical supervised learning makes use only of the initial data to predict the disease status at the follow-up, although in many cases data about medications, laboratory tests or vital signs are routinely collected about patients also at intermediate time points. This information is *privileged* (Vapnik and Vashist, 2009), as it is unavailable at the time of prediction, but can be used for training a model.

*Learning using privileged information* (LuPI) (Vapnik and Vashist, 2009), *generalized distillation* (Lopez-Paz et al., 2016) and *multi-view learning* (Rosenberg and Bartlett, 2007) have been proposed to increase the sample efficiency by leveraging privileged information in learning. Theoretical results guarantee improved learning rates (Pechyony and Vapnik, 2010; Vapnik et al., 2015) or tighter generalization bounds (Wang, 2019) for large sample sizes under appropriate assumptions. However, privileged information is not always beneficial; it must be related to the task at hand (Jonschkowski et al., 2015). Previous works do not identify such settings. Moreover, existing analyses do not state when learning with privileged information is preferable to classical learning for problems with small sample sizes—which is where efficiency is needed the most.

Karlsson et al. (2022) studied LuPI in the context of predicting an outcome observed at the end of a time series based on variables collected at the first time step. They showed that making use of data from intermediate time steps in particular settings always leads to lower or equal prediction risk—for any sample size—compared to the best unbiased model which does not make use of this privileged information. However, their method called *learning using privileged time series* (LuPTS) was limited to settings where the outcome function, and estimators of it, are linear functions of baseline features.

36th Conference on Neural Information Processing Systems (NeurIPS 2022).

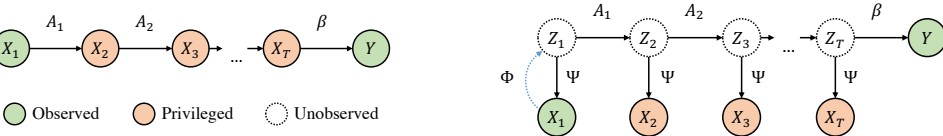

(a) Linear dynamical system, fully observed     (b) Linear latent dynamics, nonlinear observations

Figure 1: Comparison between the linear data generating process and the latent nonlinear generalization in this work. $\Psi$ indicates the observation function of the latent system, with $\Phi$ its left inverse.

Moreover, their analysis did not study how variance reduction behaves as a function of increased input dimension. Hayashi et al. (2019) also learned from privileged intermediate time points but their study was limited to empirical results for classification using generalized distillation.

There is an abundance of real-world prediction tasks with fixed follow-up times which can be framed as having access to privileged time-series information. Examples include predicting 90-day patient mortality (Karhade et al., 2019) or patient readmission in healthcare (Mortazavi et al., 2016), the churn of users of an online service over a fixed period (Huang et al., 2012) or yearly crop yields from satellite imagery of farms (You et al., 2017). In these cases, privileged time-series information comprises daily patient vitals, intermediate user interactions and daily satellite imagery respectively. We are motivated by finding sample-efficient learning algorithms that utilize privileged time-series information to improve upon classical learning alternatives in such settings.

**Contributions.** We extend the LuPTS framework to nonlinear models and prediction tasks in latent dynamical systems (Section 3). In this setting, we prove that learning with privileged information leads to lower risk when the nonlinear map connecting latent variables and observations is known up to a linear transform (Section 3.1). In doing so, we also find that when the representation dimension grows larger than the number of samples, the benefit of privileged information vanishes. We show that a privileged learner using random feature maps can learn optimal models consistently, even when the relationship between latent and observed variables is unknown, and give a practical algorithm based on this idea (Section 3.2). However, random feature methods may suffer from bias in small samples. As a remedy, we propose several representation learning algorithms aimed at trading off bias and variance (Section 3.3). In experiments, we find that privileged time-series learners with either random features or representation learning reduce variance and improve latent state recovery in small-sample settings (Section 4) on both synthetic and real-world regression tasks.

## 2 Prediction and privileged information in nonlinear time series

We aim to predict outcomes $Y$ in $\mathcal{Y} \subseteq \mathbb{R}^q$ based on covariates $X_1$ in $\mathcal{X} \subseteq \mathbb{R}^k$. $X_1$ are observed at a baseline time point $t = 1$, starting a discrete time series on the form $X_1, X_2, \ldots X_T, Y$. Outcomes are assumed to be compositions of a representation $\Phi$, a linear map $\theta$ and Gaussian noise $\epsilon$,

$$Y = h(X_1) + \epsilon, \quad \text{where} \quad h(x_1) := \theta^\top \Phi(x_1), \quad \text{and} \quad \epsilon \sim \mathcal{N}(0, \tilde{\sigma}_Y^2) . \tag{1}$$

In addition to observations of $X_1$ and $Y$, *privileged information* (PI) is available in the form of samples of random variables, $X_2, ..., X_T$, from intermediate time points between $X_1$ and $Y$, all taking values in $\mathcal{X}$. Two data generating processes (DGPs) with this structure are illustrated in Figure 1. Unlike baseline variables $X_1$, privileged information is observed *only at training time, not at test time*. Therefore, it can only benefit learning, and not inference. Data sets of $m \in \mathbb{N}_+$ training samples, $D := \{(x_{i,1}, x_{i,2}, ..., x_{i,T}, y_i)\}_{i=1}^m$, are drawn independently and identically distributed from a fixed, unknown distribution $p$ over all random variables in our system. We let $\mathbf{X}_t = [x_{1,t}, ..., x_{m,t}]^\top$ denote the data matrix of features observed at time $t = 1, ..., T$ and $\mathbf{Y} = [y_1, ..., y_m]^\top$ the vector of all outcomes observed in $D$. A learning algorithm $\mathscr{A} : \mathcal{D} \to \mathcal{H}$ maps data sets $D \in \mathcal{D}$ to hypotheses $\hat{h} \in \mathcal{H}$. An *efficient* algorithm $\mathscr{A}$ has small expected risk $\overline{R}_p(\mathscr{A})$ with respect to a loss $L : \mathcal{Y} \times \mathcal{Y} \to \mathbb{R}$ over training sets of fixed size $m$ drawn from $p$,

$$\overline{R}_p(\mathscr{A}) := \mathbb{E}_D[R_p(\hat{h})] \quad \text{where} \quad \hat{h} = \mathscr{A}(D) \quad \text{and} \quad R_p(\hat{h}) := \mathbb{E}_p[L(\hat{h}(X_1), Y)] . \tag{2}$$

We study the regression setting with $L$ the squared error, $L(y, y') = \|y - y'\|_2^2$. In analysis, we focus on univariate outcomes, $q = 1$, but all results extend to multivariate outcomes, $q > 1$.

**Algorithm 1:** Generalized LuPTS

**Input:** Data $D = (\{\mathbf{X}_t\}, \mathbf{Y})$; Repr. $\hat{\Phi}$ or kernel $\kappa$

**if** using a fixed representation $\hat{\Phi}$ **then**

$\quad \hat{\mathbf{Z}}_t = [\hat{\Phi}(x_{1,1}), ..., \hat{\Phi}(x_{m,1})]^\top$ for $t = 1, ..., T$

$\quad \hat{\theta}_{\mathrm{P}} := \big[ \prod_{t=1}^{T-1} \underbrace{(\hat{\mathbf{Z}}_t^\top \hat{\mathbf{Z}}_t)^\dagger \hat{\mathbf{Z}}_t^\top \hat{\mathbf{Z}}_{t+1}}_{\hat{A}_t} \big] \underbrace{(\hat{\mathbf{Z}}_T^\top \hat{\mathbf{Z}}_T)^\dagger \hat{\mathbf{Z}}_T^\top \mathbf{Y}}_{\hat{\beta}}$

$\quad \hat{h}_{\mathrm{P}}(\cdot) := \hat{\theta}_{\mathrm{P}}^\top \hat{\Phi}(\cdot)$

**else if** using kernel $\kappa$ **then**

$\quad \hat{\mathbf{K}}_t = \kappa(\mathbf{X}_t, \mathbf{X}_t)$ for $t = 1, ..., T$

$\quad \boldsymbol{\alpha} := \hat{\mathbf{K}}_1^\dagger \big[ \prod_{t=2}^{T} \hat{\mathbf{K}}_t \hat{\mathbf{K}}_t^\dagger \big] \mathbf{Y}$

$\quad \hat{h}_{\mathrm{P}}(\cdot) := \sum_{i=1}^{n} \alpha_i \kappa(x_{i,1}, \cdot)$

**return** $\hat{h}_{\mathrm{P}}$

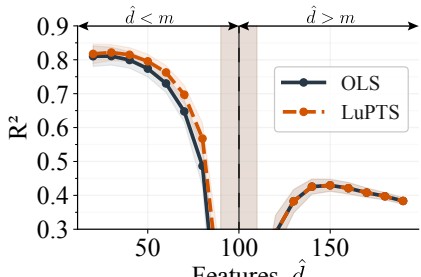

Figure 2: Two regimes of generalized LuPTS with varying feature dimension $\hat{d}$, and sample size $m = 100$. When $m > \hat{d}$, generalized LuPTS is provably never worse than OLS. When $m \leq \hat{d}$, they are equivalent. See Appendix F for details.

Our main goal is to compare *privileged learners* $\mathscr{A}_{\mathrm{P}}$, which make use of privileged information, to *classical learners* $\mathscr{A}_{\mathrm{C}}$ which learn without PI. We seek to identify conditions and algorithms for which privileged learning is more efficient, i.e., it leads to lower risk in expectation, $\overline{R}(\mathscr{A}_{\mathrm{P}}) \leq \overline{R}(\mathscr{A}_{\mathrm{C}})$ for the same number of training samples. We describe such a setting next.

### 2.1 Privileged information in latent dynamical systems

We study tasks where data are produced by a latent dynamical process. Observations $X_t$ are generated from unobserved latent states $Z_t$ through a nonlinear transformation $\Psi$, see Figure 1b. This means that the target of learning, $h(X_1) = \mathbb{E}[Y \mid X_1]$, is nonlinear in $X_1$. Latent dynamical systems like these have proven successful at modelling a variety of phenomena as for example fluid dynamics in physics (Lee and Carlberg, 2019) and human brain activity in neuroscience (Kao et al., 2015).

**Assumption 1** (Latent linear-Gaussian system). *Let $Z_1, ..., Z_T$ be latent states related as a linear-Gaussian dynamical system in a space $\tilde{\mathcal{Z}} \subseteq \mathbb{R}^d$ with $Z_1$ of arbitrary distribution. Further, let the observation function be an injective map $\Psi : \mathcal{Z} \to \mathcal{X}$ with left-inverse (representation) $\Phi : \mathcal{X} \to \mathcal{Z}$. With $A_1, ..., A_{T-1} \in \mathbb{R}^{d \times d}$, $\beta \in \mathbb{R}^d$ assume that $Z_1, ..., Z_T, X_1, ..., X_T, Y$ are generated as*

$$Z_t = A_{t-1}^\top Z_{t-1} + \epsilon_t \quad \text{for} \quad t \geq 2, \quad \text{and} \quad \forall t : X_t = \Psi(Z_t) \quad \text{and} \quad Y = \beta^\top Z_T + \epsilon_Y .$$

*where $\epsilon_t \sim \mathcal{N}(0, \sigma_t^2 I)$ and $\epsilon_Y \sim \mathcal{N}(0, \sigma_Y^2)$.*

It is easy to verify that Assumption 1 is consistent with (1), but stronger: there are more systems that map $X_1$ to $Y$ as $\theta^\top \Phi(X_1)$ than systems where additionally $X_t = \Psi(Z_t)$. Nevertheless, it is much more general than the results of Karlsson et al. (2022), limited to the linear setting, i.e. $\Phi(x_1) = x_1$.

Next, we present a generalized version of the LuPTS algorithm of Karlsson et al. (2022) that is provably preferable to classical learning when data is generated according to Assumption 1 and $\Phi$ is known up to a linear transform. Further, we discuss how a privileged learner can be made universally consistent for unknown representations $\Phi$ when combined with random features.

## 3 Efficient learning with time-series privileged information

We analyze and compare learners from the privileged and classical paradigms which produce estimates of the form $\hat{h}(x_1) = \hat{\theta}^\top \hat{\Phi}(x_1)$, as motivated by (1). We let each use representations $\hat{\Phi} : \mathcal{X} \to \hat{\mathcal{Z}} \subseteq \mathbb{R}^{\hat{d}}$ from a family $\mathcal{F}$ *shared by both paradigms*, so that the hypothesis class $\mathcal{H} \ni \hat{h}$ is shared as well.

### 3.1 Learning with true representations known up to a linear transform

In this section, we assume that privileged and classical learners have access to a common, fixed representation function $\hat{\Phi}$ which is related to the true representation function through a linear transform, meaning there exists a matrix $B$ such that $\hat{\Phi}(\cdot) = B\Phi(\cdot)$.

**Classical learning.** As a classical learner to serve as a strong comparison point for our privileged learners, we use the ordinary least-squares (OLS) linear estimator applied to the latent variables at the first time point inferred by $\hat{\Phi}$. With $\hat{\mathbf{Z}}_1 = [\hat{\Phi}(x_{1,1}), ..., \hat{\Phi}(x_{m,1})]^\top \in \mathbb{R}^{m \times \hat{d}}$,

$$\mathscr{A}_{\text{C}}(D) = \hat{h}_{\text{C}}(\cdot) := \hat{\theta}_{\text{C}}^\top \hat{\Phi}(\cdot) \quad \text{where} \quad \hat{\theta}_{\text{C}} := (\hat{\mathbf{Z}}_1^\top \hat{\mathbf{Z}}_1)^\dagger \hat{\mathbf{Z}}_1^\top \mathbf{Y} \ . \tag{3}$$

When $m \geq \hat{d}$ and $\Phi$ is known up to linear transform, $\hat{h}_{\text{C}}$ is the minimum-risk unbiased estimator of $h$ which *does not* use privileged information. To accommodate the underdetermined case where $m < \hat{d}$, the matrix inverse is replaced by the Moore-Penrose pseudo inverse $(\cdot)^\dagger$ (Penrose, 1956).

**Generalized LuPTS.** For privileged learning, we first compute $\hat{\mathbf{Z}}_t = [\hat{\Phi}(x_{1,t}), ..., \hat{\Phi}(x_{m,t})]^\top$ for $t = 1, ..., T$. Then, we independently fit parameter estimates $\hat{A}_1, ..., \hat{A}_T, \hat{\beta}$ of the dynamical system shown in Figure 1b by minimizing the squared error in single-step predictions. This equates to a series of OLS estimates in $\hat{\mathcal{Z}}$. At test time, baseline variables $x_1$ are embedded with $\hat{\Phi}$ and the latent dynamical system is simulated for $T$ time steps to form a prediction $\hat{h}_{\text{P}}(x_1) = (\hat{A}_1 \cdots \hat{A}_T \beta)^\top \hat{\Phi}(x_1)$. Putting this together, we arrive at Algorithm 1 which we call *generalized LuPTS*. We may apply a simple matrix identity and replace terms $\hat{\mathbf{Z}}_t \hat{\mathbf{Z}}_t^\top$ in Algorithm 1 by the Gram matrices $\hat{\mathbf{K}}_t = \kappa(\mathbf{X}_t, \mathbf{X}_t)$ of a reproducing kernel $\kappa$ with corresponding (implicit) feature map $\hat{\Phi}$, $\kappa(x, x') = \langle \hat{\Phi}(x), \hat{\Phi}(x') \rangle$ (Smola and Schölkopf, 1998). This variant allows for learning with unknown $\Phi$.

We now state a result which says that under Assumption 1, for an appropriate fixed representation $\hat{\Phi}$ or kernel $\kappa$, generalized LuPTS is never worse in expectation than the classical learner in (3).

**Theorem 1.** *Let $D$ be a data set of size $m$ drawn from $p$, consistent with Assumption 1. Assume that the left inverse $\Phi : \mathcal{X} \rightarrow \mathcal{Z}$ of the observation function $\Psi$ is known up to a linear transform, explicitly or through a kernel $\kappa(x, x') = \langle \hat{\Phi}(x), \hat{\Phi}(x') \rangle$, i.e., there exists a matrix $B$ with linearly independent columns such that $\hat{\Phi}(x) = B\Phi(x)$ with $\hat{\Phi} : \mathcal{X} \rightarrow \hat{\mathcal{Z}}$ for all $x \in \mathcal{X}$. Then, it holds for the privileged learner $\mathscr{A}_{\text{P}}(D) = \hat{h}_{\text{P}}$ (Algorithm 1) and the classical learner $\mathscr{A}_{\text{C}}(D) = \hat{h}_{\text{C}}$ (3),*

$$\overline{R}(\mathscr{A}_{\text{P}}) = \overline{R}(\mathscr{A}_{\text{C}}) - \mathbb{E}_{\hat{h}_{\text{P}}, X_1}[\text{Var}_D(\hat{h}_{\text{C}}(X_1) \mid \hat{h}_{\text{P}})] \ . \tag{4}$$

*Proof sketch.* First, we show that the predictions made by generalized LuPTS are invariant to a linear transform $B$ applied to $\hat{\Phi}(\cdot)$ during training, see Appendix A. Then we consider two cases: (i) When $\hat{d} \leq m$ we may re-purpose the proof of Theorem 1 in Karlsson et al. (2022), (ii) When $m < \hat{d}$ Proposition 1 below directly implies $\mathbb{E}_{\hat{h}_{\text{P}}, X_1}[\text{Var}_D(\hat{h}_{\text{C}}(X_1) \mid \hat{h}_{\text{P}})] = 0$ and $\overline{R}(\mathscr{A}_{\text{P}}) = \overline{R}(\mathscr{A}_{\text{C}})$. $\quad\square$

Theorem 1 implies that the privileged learner is at least as sample efficient as the classical one since $\text{Var}(\cdot) \geq 0$ and thus $\overline{R}(\mathscr{A}_{\text{P}}) \leq \overline{R}(\mathscr{A}_{\text{C}})$ for the same number of training samples $m$. The result is a direct generalization of the main result in Karlsson et al. (2022). We also observe that generalized LuPTS and the classical learner coincide under certain conditions when $m \leq \hat{d}$.

**Proposition 1.** *Let $\hat{\Phi} : \mathcal{X} \rightarrow \hat{\mathcal{Z}} \subseteq \mathbb{R}^{\hat{d}}$ be any map with corresponding kernel $\kappa$. Let $\hat{\mathbf{K}}_t$ be the Gram matrix of $\kappa$ applied to $\mathbf{X}_t$ and let $\hat{h}_{\text{C}}, \hat{h}_{\text{P}}$ be classical (3) and privileged (Algorithm 1) estimates. Then,*

$$\hat{\mathbf{K}}_t \text{ is invertible for all } t \implies \hat{h}_{\text{P}} = \hat{h}_{\text{C}}.$$

$\hat{\mathbf{K}}_t$ *is noninvertible whenever $m > \hat{d}$, assuming linearly independent features.*

*Proof sketch.* When the Gram matrices $\mathbf{K}_t$ are invertible, the pseudo-inverse coincides with the inverse, and factors $\mathbf{K}_t \mathbf{K}_t^\dagger$ in the LuPTS estimators cancel, making LuPTS and CL equal. $\quad\square$

**Remarks.** Theorem 1 extends the applicability of LuPTS to a) nonlinear prediction through a fixed feature map or b) kernel estimation and c) to the underdetermined case of $m < \hat{d}$. While previous work is restricted to observed linear systems our result considers the case of latent linear dynamics which only need to be identified up to a linear transform (see Figure 1). Proposition 1 does not claim that no preferable privileged learner exists; it is a statement only about generalized LuPTS. We may relate the result to the double descent characteristic previously observed for other linear estimators for fixed $m$ and varying $\hat{d}$ (Loog et al., 2020). After a phase transition around $\hat{d} = m$ LuPTS's variance reduces for a second time when it becomes equivalent to the classical learner (see Figure 2).

## 3.2 Random feature maps for unknown representations

When the true $\Phi$ is entirely unknown, as is often the case in practice, using a poor representation $\hat{\Phi}$ may yield biased results for both classical and privileged learners. A common solution in non-linear prediction is to use a universal kernel, such as the Gaussian-RBF kernel. These have dense reproducing-kernel Hilbert spaces which allow approximation of any continuous function. However, universal kernels also have positive-definite and thus invertible Gram matrices (Hofmann et al., 2008), which according to Proposition 1 eliminates any gain in sample efficiency of generalized LuPTS.

Instead, we combine our algorithm with an approximation of universal kernels—random feature maps. These methods project inputs $x$ onto $\hat{d}$ features by a random linear map $W \in \mathbb{R}^{k \times \hat{d}}$, and a nonlinear element-wise activation function. By choosing $\hat{d} < m$, we can benefit from the function approximation properties of universal kernels (see discussion below) and the variance reduction of generalized LuPTS. Popular random features include random Fourier features (RFF) (Rahimi and Recht, 2007) and random ReLU features (RRF) (Sun et al., 2018).[1]

$$\hat{\Phi}_{\text{RFF}}^\gamma(x) = \sqrt{2/\hat{d}}\left[\cos(\sqrt{2\gamma}W_\mathcal{N}^\top x + b)\right] \quad \text{and} \quad \hat{\Phi}_{\text{RRF}}^\gamma(x) = f_+(\gamma W_\mathcal{U}^\top [x; 1]) .$$

where $f_+(z) = \max(0, z)$ is the rectifier (ReLU) function and $\gamma > 0$ is a bandwidth hyperparameter.

For large enough numbers of random features and training samples, any continuous function $h$ can be approximated up to arbitrary precision by a linear map $\omega$ applied to the random features, e.g., $\hat{h}(x) = \omega^\top \hat{\Phi}_{\text{RRF}}^\gamma(x)$, see Sun et al. (2019); Rudi and Rosasco (2017). Applying the same argument to the step-wise estimators of LuPTS we can justify using random features in Algorithm 1 by the following observation: *Under appropriate assumptions, we can construct a privileged learner using random feature maps which is a universally consistent estimator of $h(x_1) = \mathbb{E}[Y|x_1]$.* The precise construction deviates somewhat from Algorithm 1, but follows the same structure. We refer to Appendix B for a precise statement. As universal consistency describes the asymptotic behaviour in the limit of infinite samples and random features, this offers only limited insight into the benefits of privileged information in small sample settings, where performance will be a bias-variance trade-off.

**Variance reduction & bias amplification.** Generalized LuPTS is only guaranteed lower variance compared to the classical estimator under Assumption 1, although our empirical results (Section 4) suggest this applies more widely. When $\hat{\Phi}$ is a bad approximation of $\Phi$, generalized LuPTS may amplify bias, increasing with the number of privileged time points, compared to classical learning. We show this theoretically in Appendix C and also empirically in Appendix F. Whether generalized LuPTS is still preferable to classical learning in terms of prediction risk appears to depend on the amount of bias that gets amplified. Our experiments imply the variance reduction mostly dominates when using random features, whereas this is not always the case for linear LuPTS. The phenomenon of bias amplification is familiar from e.g., model-based and model-free reinforcement learning (Kober et al., 2013). As the bias with random features may still be high for small sample settings, we next present privileged representation learning algorithms to trade off bias and variance more efficiently.

## 3.3 Privileged time-series representation learning

Up until now the representation $\hat{\Phi}$ was considered fixed, either because $\Phi$ was known up to a linear transform (explicitly or implicitly) or because of the use of random feature methods. Generalized LuPTS (Algorithm 1) produces minimizers $\{\hat{A}_t\}, \hat{\beta}$ of the following objective for fixed $\hat{\Phi}$,

$$\mathcal{L}_{\text{SRL}}(\hat{\Phi}, \{\hat{A}_t\}, \hat{\beta}) := \frac{1}{NT} \sum_{i=1}^N \left[ \sum_{t=1}^{T-1} \frac{1}{\hat{d}} \|\hat{A}_t^\top \hat{\Phi}(x_{i,t}) - \hat{\Phi}(x_{i,t+1})\|_2^2 + \frac{1}{q} \|\hat{\beta}^\top \hat{\Phi}(x_{i,T}) - y_i\|_2^2 \right] . \quad (5)$$

Objective (5) and the systems described by Assumption 1 lend themselves to methods which also learn the representation $\Phi$ in addition to the latent dynamics $\{A_t, \beta\}$. Next, we present three algorithms which combine the ideas of generalized LuPTS with the expressiveness of deep representation learning. All learners use equivalent encoders to represent $\hat{\Phi}(\cdot)$ and linear layers to model the relations between the latent variables $\{Z_t\}$ and the outcome $Y$. The classical learner predicts the outcome linearly from $\hat{Z}_1$. All architectures under consideration are visualized jointly in Figure 3a.

---

[1]For RFF, $[W_\mathcal{N}]_{ij} \sim \mathcal{N}(0,1)$, and for RRF, $[W_\mathcal{U}]_{ij} \sim \mathcal{U}(-1,1)$ and $b_i \sim \mathcal{U}(0, 2\pi)$.

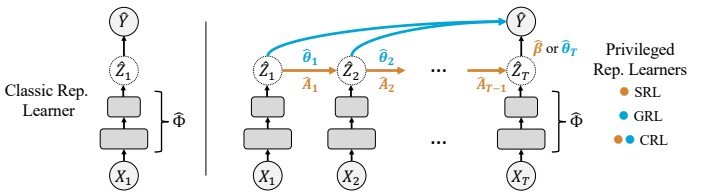 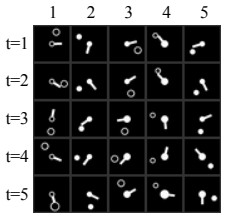

(a) Classical (left) and privileged (right) representation learners. $\hat{\Phi}$ is shared across all time steps. GRL models the direct maps $\hat{\theta}_t$ to the outcome; SRL models the single steps $\hat{A}_t$ and $\hat{\beta}$. CRL combines the two.

(b) Five example sequences from **Clocks-LGS** image data.

Figure 3: Representation learning architectures (left) and samples from **Clocks-LGS** (right).

**SRL.** The first privileged representation learner directly optimizes objective (5), just like generalized LuPTS, but now also fitting the representation $\hat{\Phi}$, parameterized by a neural network. We refer to this model as *stepwise representation learner* (SRL). As we will see in experiments, a drawback of this approach is that representations may favor predicting transitions $\hat{z}_{i,t} \to \hat{z}_{i,t+1}$ with small error, while losing information relevant for the target outcome in the process. At test time, for a new input $x_1$, SRL composes the stepwise dynamics to output $\hat{h}_{\text{P}}(x_1) = \hat{\beta}^\top \hat{A}_T^\top \ldots \hat{A}_1^\top \hat{\Phi}(x_1)$.

**CRL and GRL.** To make sure that the learned representation $\hat{\Phi}$ retains information about $Y$, we add linear outcome supervision to the representation $\hat{\Phi}(X_t)$ at each time step $t$. Recall that, by Assumption 1, the expected outcome is linear in the latent state at *any* time step. We introduce a hyperparameter $\lambda \in [0, 1]$ to trade off the two types of losses and arrive at the *combined representation learner* (CRL). With $\alpha = (\hat{\Phi}, \{\hat{A}_t\}, \{\hat{\theta}_t\})$ the entire parameter vector, CRL minimizes the objective

$$\mathcal{L}_{\text{CRL}}(\alpha) := \frac{\lambda}{NTq} \sum_{i,t} \left\| \hat{\theta}_t^\top \hat{\Phi}(x_{i,t}) - y_i \right\|_2^2 + \frac{1-\lambda}{N(T-1)\hat{d}} \sum_{i,t} \left\| \hat{A}_t^\top \hat{\Phi}(x_{i,t}) - \hat{\Phi}(x_{i,t+1}) \right\|_2^2. \quad (6)$$

We make test-time predictions using $\hat{h}_{\text{P}}(x_1) = \hat{\theta}_1^\top \hat{\Phi}(x_1)$. In experiments, we highlight the case $\lambda = 1$ where only outcome supervision is used as *greedy representation learner* (GRL). For a precise definition of the GRL objective, see Appendix D. GRL is related to multi-view learning, in which prediction of the same quantity is made from multiple "views" (cf. time points) (Zhao et al., 2017).

# 4 Experiments

We compare classical learning to variants of generalized LuPTS (Algorithm 1) and the privileged representation learners of Section 3.3 on two synthetic and two real-world time-series data sets. We (i) verify our theoretical findings by analyzing the sample efficiency and bias-variance characteristics of the given algorithms; (ii) demonstrate that generalized LuPTS with random features succeeds in settings where linear LuPTS suffers from large bias; (iii) point out that privileged representation learners offer even greater sample efficiency in practice and (iv) study how well these algorithms recover the true latent variables $\{Z_t\}$ and how this relates to predictive accuracy.

**Experimental setup.** We report the mean coefficient of determination ($R^2$), proportional to the squared-error risk $\overline{R}$, for varying sample sizes, sequence lengths and prediction horizons. Experiments are repeated and averaged over different random seeds. In each repetition, a given model performs hyperparameter tuning on the training data using random search and five-fold cross-validation before being re-trained on all training data. The test set size is 1000 samples for synthetic data and 20% of all data available for real-world data sets. We consider six privileged learners of two groups. The first group comprises generalized LuPTS with the linear kernel (LuPTS) and the two random feature maps shown in Section 3.2: Random Fourier features (Fourier RF) and random ReLU features (ReLU RF). The classical learners for this group are OLS estimators used with the same kernel or feature map. The second group consists of the representation learners SRL, CRL and GRL. For tabular data, their encoder is a multi-layer perceptron with three hidden layers of 25 neurons each. For the image data they use LeNet-5 (LeCun et al., 1989). The classical learner (Classic Rep.) uses the same encoder with a linear output layer. The results presented were found to be robust to small changes in training

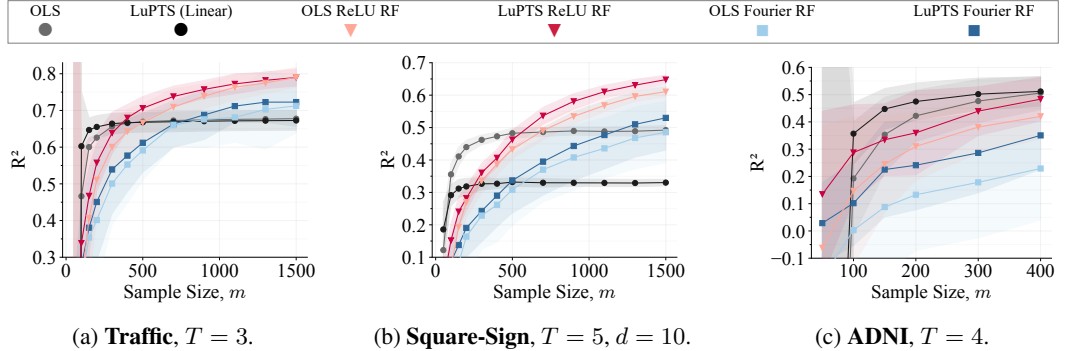

Figure 4: Predictive accuracy of generalized LuPTS on three data sets, over 60 repetitions. The shaded area represents one standard deviation above and below the mean over repetitions.

parameters such as learning rate. For details on the training process we refer to Appendix E. All experiments required less than 3000 GPU-h to complete using NVIDIA Tesla T4 GPUs.[2]

**Data sets.** We briefly describe evaluation data sets and refer to Appendix E for further details. First, we create two synthetic data sets in which latent states and outcomes are generated from linear-Gaussian systems as in Assumption 1. To produce the observations $\{X_t\}$ we use a deterministic nonlinear function $\Psi : \mathcal{Z} \to \mathcal{X}$. In the first synthetic data set, which we call **Square-Sign**, the nonlinear transformation $\Psi : \mathbb{R}^d \to \mathbb{R}^{2d}$ maps each latent feature $Z_{(t,k)}$ to a two dimensional vector such that

$$X_t \coloneqq \Psi(Z_{(t)}) = [Z_{(t,1)}^2, \mathrm{sgn}(Z_{(t,1)}), \ldots, Z_{(t,d)}^2, \mathrm{sgn}(Z_{(t,d)})]^\top.$$

The second synthetic data set uses the same latent linear system with $Z_t \in \mathbb{R}^2$ and produces square images ($28 \times 28$ pixels) as observations. As the images are reminiscent of clocks we refer to this data set as **Clocks-LGS**. Example sequences of these observations are presented in Figure 3b. The angle, size and fill of the two clock hands encode the value of the corresponding latent variable. The outcome $Y$ is a linear function of $Z_T$ with $q = 1$. For **Square-Sign**, $q = 3$.

The Metro Interstate traffic volume data set (**Traffic**) (Hogue, 2012) contains hourly records of the traffic volume on the interstate 94 between Minneapolis and St. Paul, MN. In addition, the data contains weather features and a holiday indication. We predict the traffic volume for a fixed time horizon given the present observations. Privileged information is observed every four hours.

We also predict the progression of Alzheimer's disease (AD) as measured by the outcome of the Mini Mental State Examination (MMSE) (Galea and Woodward, 2005). The anonymized data were obtained through the Alzheimer's Disease Neuroimaging Initiative (**ADNI**) (ADNI, 2004). The outcome of interest is the MMSE score 48 months after the first examination. Privileged information are the measurements at 12, 24 and 36 months. In addition we tested our algorithms on the $\mathbf{PM}_{2.5}$ data set (Liang et al., 2016), where we predict the air quality in five Chinese cities (see Appendix F).

## 4.1 Sample efficiency, bias and variance

The main goal of our work is to improve learning efficiency by incorporating privileged time-series information. Across almost all prediction tasks and sample sizes, nonlinear variants of generalized LuPTS outperform their classical counterpart in terms of sample efficiency as can be seen in Figure 4. On the **Traffic** prediction task, LuPTS ReLU RF outperforms linear LuPTS as the former appears to exhibit less bias. On the synthetic data of **Square-Sign**, linear models reach their best accuracy quickly, while they are limited by their lack of expressiveness. Generalized LuPTS amplifies this bias, making linear LuPTS worse than OLS. Random feature methods attain higher accuracy here but are generally less sample efficient. Generalized LuPTS combined with random features manages to decrease this gap significantly. On **ADNI**, we don't see a benefit of using nonlinear models in general. We make similar observations on the $\mathbf{PM}_{2.5}$ air quality data set, see Appendix F.

---

[2]Code to reproduce all results is available at https://github.com/Healthy-AI/glupts.

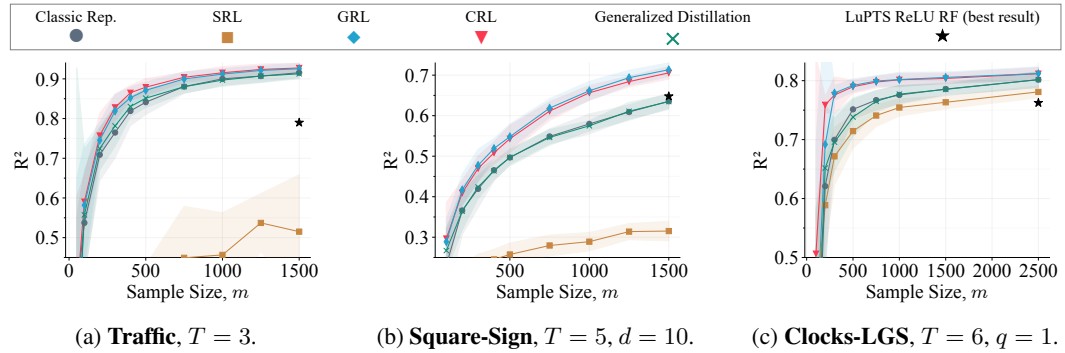

Figure 5: Predictive accuracy of the representation learners over 25 repetitions. Generalized distillation is used as proposed by Hayashi et al. (2019). For details, see Appendix E.

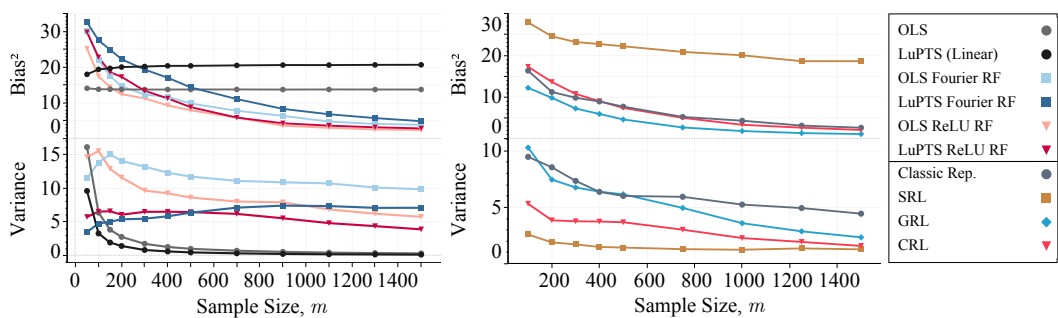

Figure 6: Estimated squared bias and variance of models trained on **Square-Sign** ($T = 5$, $d = 10$, $q = 3$) over random training sets (60 for left group, 25 for the right), evaluate on 1000 test points.

The representation learners proposed in Section 3.3 are evaluated on the same data sets and on the image prediction task **Clocks-LGS**. Example results are displayed in Figure 5. These demonstrate that directly transferring the LuPTS objective to neural networks in the form of SRL results in subpar performance. SRL does not seem to have a strong enough incentive to learn representations which accurately predict the outcome. Generalized distillation for privileged time series as suggested by Hayashi et al. (2019), does not improve upon classical learning in our tasks. On all experiments displayed in Figure 5, CRL and GRL outperform the classical learner. The predictive accuracy of these models is similar on most tasks, as may be explained by the fact that CRL may reduce to GRL when choosing $\lambda = 1$ in objective 6. Noticeably, the general observation of GRL and CRL being more sample efficient than the classic model, neither appears to depend on the neural architecture used for the encoder, nor does the modality of the data play an important role, as the image prediction task **Clocks-LGS** (Figure 5c) demonstrates. For additional empirical results, we refer to Appendix F.

To analyze the bias and variance characteristics of our algorithms, we can estimate the expected squared prediction *bias*, $\mathbb{E}_{X_1}[(\mathbb{E}_D[\mathscr{A}(D)](X_1) - \mathbb{E}[Y|X_1])^2]$, by computing $\mathbb{E}_Y[Y|X_1]$ in synthetic DGPs in addition to the variance of the different estimators. Figure 6 depicts bias and variance for all models on the **Square-Sign** data. On the left panel, all variants of generalized LuPTS exhibit lower variance than classical learning, despite being biased. This holds generally: *Across all experiments, we never encountered an example where the use of privileged information has not resulted in lower variance compared to classical learning.* On the contrary, the privileged learners suffer higher bias than the comparable classical learners because of the bias compounding over the individual prediction steps as shown in Appendix C. For the random feature variants however, the bias decreases with the number of samples. The representation learners display similar characteristics on the right panel of Figure 6. Learning the transitions between latent variables $Z_t$ appears to be associated with low variance and high bias as demonstrated by the results of SRL. GRL however, which does not model these transitions, exhibits the lowest bias and the largest variance of all privileged learners. As CRL is able to trade off between these two objectives, the estimates for its variance and bias lie in between the corresponding values of the other two privileged learners.

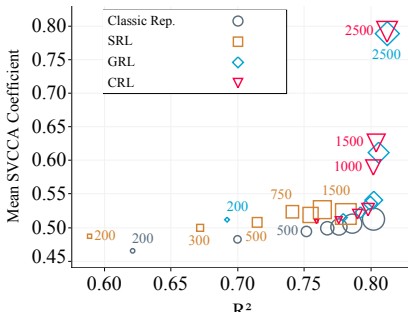
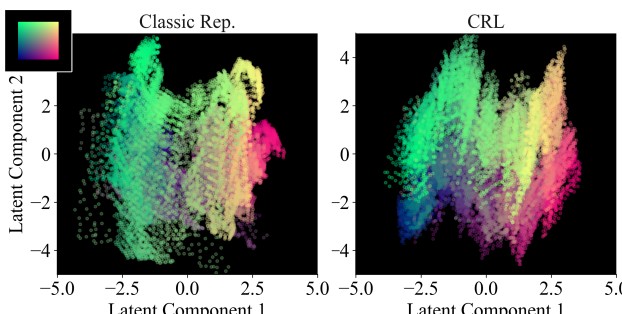

(a) Mean $R^2$ and SVVCA coefficient over 25 training runs. Markers represent average results over repeated experiments with size indicating the sample size, $m$.

(b) Visualizing the representations learned by Classic Rep. and CRL after applying SVCCA. The recovery target is shown in the top left corner. Both estimators are best in class and trained on 2500 samples.

Figure 7: Analyzing the representations learned by the models of Section 3.3 using SVVCA on the **Clocks-LGS** image regression data set (same setup as in Figure 5c).

## 4.2 Latent variable recovery

Under Assumption 1, it is sufficient to identify the representation function $\Phi$ up to a linear transform $B$ to have provable gains from privileged information over a classical learner. In synthetic data, we can assess to what extent a representation $\hat{\Phi}$ with this property has been found. As a proxy for the existence of such a transform, we can compute a measure of correlation between $\hat{\Phi}(X)$ and $\Phi(X)$, such as the Canonical Correlation Analysis (CCA) (Hotelling, 1936). Raghu et al. (2017) introduced a modified version called Singular Vector Canonical Correlation Analysis (SVCCA) to compute correlations when dealing with noisy dimensions in neural network representations.

The mean SVCCA coefficients $\bar{\rho}$ and predictive accuracy of the representation learners are visualized in Figure 7a. One notices that GRL and CRL produce higher correlation coefficients than the classical learner while also predicting the outcome on the **Clocks-LGS** task more accurately. Further comparing the representations learned by privileged and classical models, Figure 7b shows a visual example of the improved latent recovery of CRL on the same task. Concluding, the use of privileged time-series information does not only increase the sample efficiency of existing algorithms but can also aid the recovery of latent variables in latent dynamical systems.

## 5 Related work

Existing analyses for learning with privileged information guarantee improvements in asymptotic sample efficiency under strong assumptions (Vapnik and Vashist, 2009) but are insufficient to establish a clear preference for LuPI learners for a fixed sample size. For example, Pechyony and Vapnik (2010) showed that for a specialized problem construction, empirical risk minimization (ERM) using privileged information can achieve fast learning rates, $\mathcal{O}(1/n)$, while classical (non-privileged) ERM can only achieve slow rates, $\mathcal{O}(1/\sqrt{n})$. We are not aware of any generalization theory tight enough to establish a lower bound on the risk of a classical learner larger than an upper bound for a PI learner. Our problem is also related to multi-task (representation) learning (Maurer et al., 2016), see especially (5)–(6). However, our goal is different in that only a single task is of interest after learning.

In estimation of causal effects, learning from *surrogate outcomes* (Prentice, 1989) has been proposed as a way to increase sample efficiency. Surrogates are variables related to the outcome which may be available even when the outcome is not (Athey et al., 2019). We can view these as privileged information. While the problem shares structure with ours, the goal is to compensate for *missing* outcomes and analytical results give no guarantees for improved efficiency when both surrogates outcomes are always observed (Chen et al., 2008; Kallus and Mao, 2020). Guo and Perković (2022) showed that, in the context of a linear-Gaussian system on a directed acyclic graph, a recursive least-squares estimator is the asymptotically most efficient estimator of causal effects using only the sample covariance. In the linear case on a path graph, their estimator coincides with ours. However, no analysis is provided for the nonlinear case or for fixed sample sizes.

# 6 Discussion

We have presented learning algorithms for predicting nonlinear outcomes by utilizing time-series privileged information during training. We prove that our estimator is preferable to classical learning when data is generated from a latent-variable dynamical system with partially known components. The proof holds for the case when the latent dynamical system is recovered by the representation function up to a linear transform, assuming that a left inverse of the true observation generating function exists. However, this assumption does not appear to be necessary, as our empirical results demonstrate that privileged learning is preferable to classical learning even when these conditions cannot be guaranteed. Consequently, a more general theoretical result where less is known about the latent system seems attainable. For example, one might consider a case in which only a few independent components of the latent variables are recovered by the representation used by the learning algorithm, while other components are treated as noise.

When the latent dynamical system is entirely unknown, we create practical estimators using random feature embeddings which outperform the corresponding classical learner across experiments. We show that a universally consistent learner can be constructed based on this idea, with slightly different form. As a further alternative, we propose representation learning methods of related form using neural networks and demonstrate the empirical benefits also of this estimator over classical learning. In experiments, we analyze how the gap in risk between privileged and classical learning changes for different prediction horizons, as displayed in Figure 16 in Appendix F. The results suggest that the risk advantage of privileged learners grows with the sequence length of the prediction task, despite the fact that the task becomes more difficult at the same time.

Our work focuses on the setting where data is observed as a time series. This setting is chosen for its simple causal structure given by (latent) linear-Gaussian systems and because it can be motivated from many different applications. However, the ideas presented are not specifically tied to time and also apply in the case when all variables are observed simultaneously as long as the causal structure remains sequential. Moreover, we believe that the theory presented here is not limited to sequential settings and generalizes to other causal structures, in particular directed acyclic graphs that connect the baseline covariates to the outcome. In either case, one might only have access to the baseline variable at test time. For example, the timely collection of data for all covariates at test time might be very expensive or even impossible.

As pointed out before, Theorem 1 requires the recovery of the latent variables up to a linear transformation. Nonlinear independent component analysis (ICA) (Hyvarinen and Morioka, 2016) aims to solve precisely this problem and has been applied to time series via time-contrastive learning. This makes for an interesting connection between learning using privileged information and nonlinear ICA, as the experiments of Section 4.2 suggest that privileged time-series information aids the recovery of latent variables. Other remaining challenges include providing risk guarantees for learning with biased representations (including deep neural networks), with regularized estimators, and for more general data generating processes with weaker structural assumptions. We are hopeful that the utility demonstrated in this work will inspire future research to overcome these limitations.

## Acknowledgments and Disclosure of Funding

We would like to thank Anton Matsson and Rickard Karlsson for insightful feedback and the Alzheimer's Neuroimaging Initiative (ADNI) for collecting and providing the data on Alzheimer's disease used in this project. The present work was funded in part by the Wallenberg AI, Autonomous Systems and Software Program (WASP) funded by the Knut and Alice Wallenberg Foundation.

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
