$$\overline{R}(\mathscr{A}_{\mathrm{P}}) = \overline{R}(\mathscr{A}_{\mathrm{C}}) - \mathbb{E}_{\hat{h}_{\mathrm{P}}, X_1}[\mathrm{Var}_D(\hat{h}_{\mathrm{C}}(X_1) \mid \hat{h}_{\mathrm{P}})] . \tag{4}$$

*Proof sketch.* First, we show that the predictions made by generalized LuPTS are invariant to a linear transform $B$ applied to $\hat{\Phi}(\cdot)$ during training, see Appendix A. Then we consider two cases: (i) When $\hat{d} \leq m$ we may re-purpose the proof of Theorem 1 in Karlsson et al. (2022), (ii) When $m < \hat{d}$ Proposition 1 below directly implies $\mathbb{E}_{\hat{h}_{\mathrm{P}}, X_1}[\mathrm{Var}_D(\hat{h}_{\mathrm{C}}(X_1) \mid \hat{h}_{\mathrm{P}})] = 0$ and $\overline{R}(\mathscr{A}_{\mathrm{P}}) = \overline{R}(\mathscr{A}_{\mathrm{C}})$. $\

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

# Appendix

## A Proof of Theorem 1

Our proof requires an additional technical assumption: that the matrix of true latent states $\mathbf{Z}_t = [\Phi(x_{1,t}), ..., \Phi(x_{m,t})]^\top$, for all $t$, for a random data set $D$ has independent columns with probability 1. This implies that $\text{rank}(\mathbf{Z}_t) = d$ and $m \geq d$. We consider this a minor restriction since it would only be violated if either a) two or more components of $Z_t$ were perfectly correlated—in this case, a smaller system with same distributions over observations could always be constructed—or b) if we observe fewer samples than necesary to determine the system ($m < d$). Note that this *does not* require that the dimension $\hat{d}$ of the estimated representation $\hat{\Phi}$ is smaller than $m$. We begin by proving that both classical and LuPTS estimators are invariant to a particular form of linear transformation of the representation $\hat{\Phi}$.

**Lemma 1.** *Assume we have a latent linear Gaussian system as defined in Assumption 1 such that for a data set $D$ of $m$ samples, the matrix of true latent states $\mathbf{Z}_t = [\Phi(X_{1,t}); ...; \Phi(X_{m,t})]$ has linearly independent columns with probability 1. Let $\mathscr{A}_P^\Phi$ be the LuPTS algorithm using the system's true map $\Phi(\cdot)$ with $\Phi(\Psi(z)) = z \ \forall z \in \mathcal{Z}$. Let $\mathscr{A}_P^{\hat{\Phi}}$ be the same algorithm using a different map $\hat{\Phi}(\cdot)$. We assume that $\exists B : \hat{\Phi}(x) = B\Phi(x) \forall x \in \mathcal{X}$. Analogously, we denote the classical learners $\mathscr{A}_C^{\hat{\Phi}}$ and $\mathscr{A}_C^\Phi$. If $B \in \mathbb{R}^{\hat{d} \times d}$ has linearly independent columns we have*

$$\hat{h}_P^{\hat{\Phi}}(x) = \hat{h}_P^\Phi(x)$$

$$\text{and } \hat{h}_C^{\hat{\Phi}}(x) = \hat{h}_C^\Phi(x).$$

*Proof.* Let $\mathbf{Z}_t \in \mathbb{R}^{m \times d}$ be made up of the rows $\mathbf{Z}_{t(i,:)} = \Phi(\mathbf{X}_{t(i,:)})$ when $\mathbf{X}_t \in \mathbb{R}^{m \times k}$ is the design matrix belonging to data set $D$. In the same fashion we define $\hat{\mathbf{Z}}_t \in \mathbb{R}^{m \times \hat{d}}$ using the map $\hat{\Phi}$ instead. By assumption, $\mathbf{Z}_t$ and $B \in \mathbb{R}^{\hat{d} \times d}$ have independent columns such that $B^\dagger B = I$. These assumptions are used for matrix identities involving the Moore-Penrose inverse below. We compute the prediction on a new test point $x$ for the classical learner:

$$
\begin{aligned}
\hat{h}_C^{\hat{\Phi}}(x) &= \left( (\hat{\mathbf{Z}}_1^\top \hat{\mathbf{Z}}_1)^\dagger \hat{\mathbf{Z}}_1^\top Y \right)^\top \hat{\Phi}(x) \\
&= \left( \left( (\mathbf{Z}_1 B^\top)^\top (\mathbf{Z}_1 B^\top) \right)^\dagger (\mathbf{Z}_1 B^\top)^\top \mathbf{Y} \right)^\top B\Phi(x) \\
&= \left( (\mathbf{Z}_1 B^\top)^\dagger (B\mathbf{Z}_1^\top)^\dagger (\mathbf{Z}_1 B^\top)^\top \mathbf{Y} \right)^\top B\Phi(x) \\
&= \left( (B^\top)^\dagger \mathbf{Z}_1^\dagger (\mathbf{Z}_1^\top)^\dagger B^\dagger (\mathbf{Z}_1 B^\top)^\top \mathbf{Y} \right)^\top B\Phi(x) \\
&= \left( (B^\top)^\dagger (\mathbf{Z}_1^\top \mathbf{Z}_1)^\dagger \underbrace{B^\dagger B}_{=I} \mathbf{Z}_1^\top \mathbf{Y} \right)^\top B\Phi(x) \\
&= \left( (\mathbf{Z}_1^\top \mathbf{Z}_1)^\dagger \mathbf{Z}_1^\top \mathbf{Y} \right)^\top \underbrace{B^\dagger B}_{=I} \Phi(x) \\
&= \hat{h}_C^\Phi(x)
\end{aligned}
$$

The arguments for the privileged learners are analogous:

$$
\begin{aligned}
\hat{h}_P^{\hat{\Phi}}(x) &= \left[ (\hat{\mathbf{Z}}_1^\top \hat{\mathbf{Z}}_1)^\dagger \hat{\mathbf{Z}}_1^\top \hat{\mathbf{Z}}_2 \, (\hat{\mathbf{Z}}_2^\top \hat{\mathbf{Z}}_2)^\dagger \hat{\mathbf{Z}}_2^\top \hat{\mathbf{Z}}_3 \, \ldots \, (\hat{\mathbf{Z}}_T^\top \hat{\mathbf{Z}}_T)^\dagger \hat{\mathbf{Z}}_T^\top \mathbf{Y} \right]^\top \hat{\Phi}(x) \\
&= \left[ (B^\top)^\dagger (\mathbf{Z}_1^\top \mathbf{Z}_1)^\dagger B^\dagger B \mathbf{Z}_1^\top \mathbf{Z}_2 B^\top (B^\top)^\dagger (\mathbf{Z}_2^\top \mathbf{Z}_2)^\dagger B^\dagger B \mathbf{Z}_2^\top \mathbf{Z}_3 B^\top \right.
\end{aligned}
$$

$$\dots \ (B^\top)^\dagger (\mathbf{Z}_T^\top \mathbf{Z}_T)^\dagger B^\dagger B \mathbf{Z}_T^\top \mathbf{Y} \Big]^\top B \Phi(x)$$

$$= \Big[ (\mathbf{Z}_1^\top \mathbf{Z}_1)^\dagger \mathbf{Z}_1^\top \mathbf{Z}_2 (\mathbf{Z}_2^\top \mathbf{Z}_2)^\dagger \mathbf{Z}_2^\top \mathbf{Z}_3 (\mathbf{Z}_T^\top \mathbf{Z}_T)^\dagger \mathbf{Z}_T^\top \mathbf{Y} \Big]^\top B^\dagger B \Phi(x)$$

$$= \hat{h}_{\mathrm{P}}^{\Phi}(x)$$

$\square$

**Proof of Theorem 1.** We consider the generalized LuPTS estimator $\hat{h}_{\mathrm{P}}^{\hat{\Phi}}(\cdot)$ treating different cases for the number of samples $m$ and latent state dimension $d$ in turn. By the added technical assumption, that the true latent state $\mathbf{Z}_t = [\Phi(x_{1,t}), ..., \Phi(x_{m,t})]^\top$ has rank $d$ for all $t$ with probability 1, and the assumption that $\hat{\Phi}$ is linearly close to $\Phi$, by a matrix $B \in \mathbb{R}^{\hat{d} \times d}$ of rank $d$ such that $\hat{\mathbf{Z}}_t = \mathbf{Z}_t B^\top$, we get that $\mathrm{rank}(\hat{\mathbf{Z}}_t) = d$ for all $t$. This also implies $d \leq \hat{d}$.

(i) $m = d$: In this case, the Gram matrix $\mathbf{K}_t = \hat{\mathbf{Z}}_t \hat{\mathbf{Z}}_t^\top$ has full rank and thus is invertible for all $t$. By Proposition 1, LuPTS coincides with the classical learner. Hence, $\mathbb{E}_{\hat{h}_{\mathrm{P}}, X_1}[\mathrm{Var}_D(\hat{h}_{\mathrm{C}}(X_1) \mid \hat{h}_{\mathrm{P}})] = 0$ and $\overline{R}(\mathscr{A}_{\mathrm{P}}) = \overline{R}(\mathscr{A}_{\mathrm{C}})$.

(ii) $m < d$: In this case, there does not exists a linearly close, as defined above, representation $\hat{\Phi}$ to $\Phi$ since the rank of $\hat{\mathbf{Z}}_t$ must be smaller than $d$. This contradicts that $\mathrm{rank}(\hat{\mathbf{Z}}_t) = d$. Independently, if the conditions of Proposition 1 hold, the same equivalence holds as in the case $m = d$.

(iii) $m > d$: In this case, the kernel Gram matrix $\mathbf{K}_t = \hat{\mathbf{Z}}_t \hat{\mathbf{Z}}_t^\top$ has rank $d < m$ and is never invertible.

Three sub-cases remain: a) When $\hat{d} = d$, the matrix $B$ is invertible and square, the covariance matrix $\hat{\Sigma}_t = \hat{\mathbf{Z}}_t^\top \hat{\mathbf{Z}}_t$ is invertible for all $t$ and our estimator coincides with linear LuPTS (Karlsson et al., 2022) in the space implied by $\hat{\Phi}$. To see this, note that Lemma 1 implies that $\hat{h}_{\mathrm{P}}^{\hat{\Phi}}(\cdot)$ makes the same predictions as a different generalized LuPTS estimator $\hat{h}_{\mathrm{P}}^{\Phi}(\cdot)$ using the true map $\Phi$ when the two representation functions are related through $B$, as defined in the Theorem statement. Consequently, we may analyze the latter estimator instead of the first. It uses the parameter

$$\hat{\theta}_{\mathrm{P}} := \Bigg[ \prod_{t=1}^{T-1} \underbrace{(\mathbf{Z}_t^\top \mathbf{Z}_t)^\dagger \mathbf{Z}_t^\top \mathbf{Z}_{t+1}}_{\hat{A}_t} \Bigg] \underbrace{(\mathbf{Z}_T^\top \mathbf{Z}_T)^\dagger \mathbf{Z}_T^\top \mathbf{Y}}_{\hat{\beta}} \ .$$

We know by assumption that the covariance matrices $\Sigma_t = (\mathbf{Z}_t^\top \mathbf{Z}_t) \in \mathbb{R}^{d \times d}$ have full rank for all $t$. This implies that the Moore-Penrose pseudoinverse $(\cdot)^\dagger$ may be replaced by the regular matrix inverse $(\cdot)^{-1}$ in the expression above, yielding

$$\hat{\theta}_{\mathrm{P}} = \Bigg[ \prod_{t=1}^{T-1} \underbrace{(\mathbf{Z}_t^\top \mathbf{Z}_t)^{-1} \mathbf{Z}_t^\top \mathbf{Z}_{t+1}}_{\hat{A}_t} \Bigg] \underbrace{(\mathbf{Z}_T^\top \mathbf{Z}_T)^{-1} \mathbf{Z}_T^\top \mathbf{Y}}_{\hat{\beta}}$$

which is equivalent to the LuPTS estimator of Karlsson et al. (2022) used on a linear-Gaussian system in the space $\mathcal{Z}$ rather than in $\mathcal{X}$. In this case, Theorem 1 from Karlsson et al. (2022) yields the desired result. b) If $\hat{d} > d$, $\hat{\Sigma}$ is not invertible but, due to Lemma 1, we can instead study a representation which is an appropriate linear transform $B \in \mathbb{R}^{\hat{d} \times d}$ away from $\hat{\mathbf{Z}}$, and apply the Karlsson et al. (2022) result as described for the case $\hat{d} = d$. Note that in this case $B$ is non-square but has linearly independent columns as required. c) If $\hat{d} < d$, the assumed matrix $B$ cannot exist with the stated conditions (the assumptions of Theorem 1 are not satisfied).

$\square$

# B Universality of random features

A learning algorithm $\mathscr{A}$ is said to be universally consistent if, for any continuous function $h$, the output of $\mathscr{A}$ converges in probability to $h$. That is, for a random dataset $D_m$ of $m$ i.i.d. samples

drawn from a distribution $p$, and any $\epsilon > 0$,

$$\lim_{m \to \infty} \Pr[\|\mathscr{A}(D_m) - h\|_{L^2(p)} > \epsilon] = 0 .$$

Sun et al. (2019) prove that (norm-bounded) linear regression applied to random ReLU features (RRF) is universally consistent. Specifically, for any $\epsilon, \delta > 0$, there is a finite number of random features $\hat{d}$ and samples $m$, such that the estimator

$$\hat{h}_{\mathrm{RRF}}(x) = \hat{\theta}^\top \hat{\Phi}_{\mathrm{RRF}}^{\gamma,\hat{d}}(x) \quad \text{with} \quad \hat{\theta} = \underset{\theta : \|\theta\|_2^2 < R^2}{\arg\min} \frac{1}{m} \sum_{i=1}^m (\theta^\top \hat{\Phi}_{\mathrm{RRF}}^{\gamma,\hat{d}}(x_i) - y_i)^2$$

achieves an error of at most $\epsilon$ with probability $\geq 1 - \delta$ for univariate continuous functions of $x$. Universal consistency (as $\hat{d} \to \infty, m > \hat{d}$) follows as a result. We can apply the same idea to a version of generalized LuPTS by considering each parameter estimate of the latent dynamical system given $\hat{\Phi}$, with norms restricted by $R$. We drop the subscript RRF from $\hat{\Phi}$ moving forward, and continue to use random ReLU features. For the final prediction step of $Y$, we let

$$\hat{\beta} = \underset{b : \|b\|_2^2 < R^2}{\arg\min} \frac{1}{m} \sum_{i=1}^m (b^\top \hat{\Phi}^{\gamma,\hat{d}_T}(x_{i,T}) - y_i)^2 \tag{7}$$

Progressing recursively backward from $t = T$, we let

$$[\hat{A}_t]_{j,:} = \underset{a : \|a\|_2^2 < R^2}{\arg\min} \frac{1}{m} \sum_{i=1}^m (a^\top \hat{\Phi}^{\gamma,\hat{d}_t}(x_{i,t}) - \hat{\Phi}^{\gamma,\hat{d}_{t+1}}(x_{i,t+1})_j)^2 \quad \text{for } j \in [\hat{d}_{t+1}], t \in [T-1] \tag{8}$$

where $\hat{\Phi}^{\gamma,\hat{d}_t}$ are random features specific to $t$. Since the target of each prediction at $t + 1$ is fixed with respect to the features used as input at $t$, the result from Sun et al. (2019) can be used to give a learning guarantee for each step. The construction differs from the standard generalized LuPTS formulation as $\hat{\Phi}$ is not shared between time steps, and so $\hat{A}_t$ will be non-square in general. We believe that this is merely a limitation of the proof technique and that the results hold for shared random features and square transitions.

Let $\mathcal{H}_{\mathrm{ReLU}}$ denote the set of linear functions applied to $\hat{d}$ random ReLU features, with uniform random projection coefficients $\omega_j \sim \mathcal{U}(\mathbb{S}^d), j = 1, ..., \hat{d}$.

$$\mathcal{H}_{\mathrm{ReLU},\hat{d}} = \left\{ h : \mathcal{X} \to \mathbb{R}, h(x) = \sum_{j=1}^{\hat{d}} a_j \sigma(\omega_j^\top [x; 1]) \right\} .$$

**Corollary 1** (Follows from Proposition 5 in Sun et al. (2019)). *Let Assumption 1 hold with noiseless transitions and outcomes, $\epsilon_t = 0$ for $t = 2, ..., T$, and $\epsilon_Y = 0$. Define $g_t^*(x_t) := \Psi(A_t^\top \Phi(x_t)) = x_{t+1}$ and assume that for any fixed RRF representation $\hat{\Phi}^{\gamma,\hat{d}_{t+1}}$, each component $j = 1, ..., \hat{d}_{t+1}$ of the transition target satisfies $\hat{\Phi}^{\gamma,\hat{d}_{t+1}}(g_t^*(\cdot))(j) \in \mathcal{H}_{\mathrm{ReLU},\hat{d}_t}$. Let $\hat{A}_t$ be the minimizer of the single-step transitions as defined in (8). Then, for any $\delta > 0, \epsilon > 0$ there is a number of random features $\hat{d} = \hat{d}(\epsilon, \delta)$ and samples $m = m(\epsilon, \delta)$, such that with probability $\geq 1 - \delta$,*

$$\|\hat{A}_t^\top \hat{\Phi}^{\gamma,\hat{d}_t}(x_t) - \hat{\Phi}^{\gamma,\hat{d}_{t+1}}(x_{t+1})\| \leq \epsilon .$$

The result follows from Proposition 5 in Sun et al. (2019) applied to the transition functions in our problem. Putting this together for all time-steps, we get the following result.

**Proposition 2** (Universal consistency of RRF privileged learner). *Let Assumption 1 hold with $\epsilon_t = 0, \epsilon_Y = 0$. By Corollary1 and Sun et al. (2019), we have for any $\delta > 0$, $\epsilon > 0, \gamma > 0$ and a sequence $(\hat{d}_1, ..., \hat{d}_T)$ of sufficiently large numbers of random features and samples $m$, that with probability at least $1 - \delta/T$, for the privileged estimator defined in (8), (7),*

$$\|\hat{\beta}^\top \hat{\Phi}^{\gamma,\hat{d}_T}(X_T) - Y\|_{L^2(p)} \leq \epsilon$$

*and*

$$\forall t = 1, ..., T - 1 : \|\hat{A}_t^\top \hat{\Phi}^{\gamma,\hat{d}_t}(X_t) - \hat{\Phi}^{\gamma,\hat{d}_{t+1}}(X_{t+1})\|_{L^2(p)} \leq \epsilon,$$

*Then, further assume that the largest eigenvalue $\lambda_{\max}(\hat{A}_t) \leq C$ for any $t$ and $\|\hat{\beta}\| \leq C$. Then, with probability at least $1 - \delta$,*

$$\|(\hat{A}_1 \cdots \hat{A}_{T-1} \hat{\beta})^\top \hat{\Phi}^{\gamma,\hat{d}_1}(X_1) - Y\|_{L^2(p)} \leq TC^T \epsilon$$

*Proof.* Let $\|\cdot\| = \|\cdot\|_{L^2(p)}$. Then, letting $\hat{\Phi}^t = \hat{\Phi}^{\gamma,\hat{d}_t}$, and applying a union bound to each of the $T$ $(\epsilon,\delta)$-assumptions, and a series of Cauchy-Schwarz inequalities,

$$\|(\hat{A}_1 \cdots \hat{A}_{T-1}\hat{\beta})^\top \hat{\Phi}^1(X_1) - Y\|$$
$$= \|(\hat{A}_1 \cdots \hat{A}_{T-1}\hat{\beta})^\top \hat{\Phi}^1(X_1) - \hat{\beta}^\top \hat{\Phi}^T(X_T) + \hat{\beta}^\top \hat{\Phi}^T(X_T) - Y\|$$
$$\leq \|(\hat{A}_1 \cdots \hat{A}_{T-1}\hat{\beta})^\top \hat{\Phi}^1(X_1) - \hat{\beta}^\top \hat{\Phi}^T(X_T)\| + \underbrace{\|\hat{\beta}^\top \hat{\Phi}^T(X_T) - Y\|}_{\leq \epsilon}$$
$$\leq \|(\hat{A}_1 \cdots \hat{A}_{T-1}\hat{\beta})^\top \hat{\Phi}^1(X_1) - \hat{\beta}^\top \hat{A}_{T-1}^\top \hat{\Phi}^{T-1}(X_{T-1})$$
$$+ \hat{\beta}^\top (\hat{A}_{T-1}^\top \hat{\Phi}^{T-1}(X_{T-1}) - \hat{\Phi}^T(X_T))\| + \epsilon$$
$$\leq \|(\hat{A}_1 \cdots \hat{A}_{T-1}\hat{\beta})^\top \hat{\Phi}^1(X_1) - (\hat{A}_{T-1}\hat{\beta})^\top \hat{\Phi}^{T-1}(X_{T-1})\|$$
$$+ \underbrace{\|\hat{\beta}^\top (\hat{A}_{T-1}^\top \hat{\Phi}^{T-1}(X_{T-1}) - \hat{\Phi}^T(X_T))\|}_{\leq C\epsilon} + \epsilon$$
$$...$$
$$\leq TC^T\epsilon .$$

$\square$

**Remark 1.** Proposition 2 shows that a privileged learner with random ReLU features can be turned into a universally consistent estimator of any (noiseless) continuous function of $X_1$ by letting each time step have its own random feature representation of appropriate size and adding a norm constraint to each linear transformation. The construction in (8), (7) deviates from Algorithm 1 primarily in that the random feature representations used at each time step are different, but the overall structure is maintained: At training, predictions of $Y$ are made from an embedding $\hat{Z}_T = \hat{\Phi}^T(X_T)$ of $X_T$, predictions of $\hat{Z}_T$ are made from an embedding $\hat{Z}_{T-1} = \hat{\Phi}^{T-1}(X_{T-1})$, and so on. At test time, the transition matrices $\hat{A}_1, ..., \hat{A}_T, \hat{\beta}$ are multiplied and applied to $\hat{Z}_1 = \hat{\Phi}^1(X_1)$. We conjecture that a similar argument can be applied to the construction in Algorithm 1 by letting the dimension of each time step approach $\infty$.

## C  Compounding bias

We can describe the compounding bias of the LuPTS estimator due to a biased representation $\hat{\Phi}$, in comparison with the standard OLS estimator, by propagating the error in $\hat{\Phi}$ through the estimates. Assume that
$$Y = \theta^\top \Phi(X_1) + \epsilon = (A_1 \cdots A_{T-1}\beta)^\top \Phi(X_1) + \epsilon'$$

Then, let $Z_t = \Phi(X_t)$ and for an estimate $\hat{\Phi}$, assumed for simplicity to have the same dimension, $\hat{d} = d$,
$$\hat{Z}_t = \hat{\Phi}(X_t) = \Phi(X_t) + R_t$$

where $R_t$ is the residual w.r.t. $\Phi$. Let bold-face variables indicate multi-sample equivalents of all variables. Further, define $\Sigma_t = \mathbf{Z}_t^\top \mathbf{Z}_t$ and $\hat{\Sigma}_t = \hat{\mathbf{Z}}_t^\top \hat{\mathbf{Z}}_t$.

Fitting $\hat{\theta}$ to $\hat{\Phi}$ using the classical learner (OLS) yields an estimate
$$\hat{\theta}_c = \hat{\Sigma}_1^{-1}\hat{\mathbf{Z}}_1^\top \mathbf{Y}$$

Now, define $\Omega_t = \mathbf{R}_t^\top \hat{\mathbf{Z}}_t + \mathbf{Z}_t^\top \mathbf{R}_t$ and we have
$$\hat{\theta}_c = (\Sigma_1 + \Omega_1)^{-1}(\mathbf{Z}_1 + \mathbf{R}_1)^\top \mathbf{Y}$$
$$= (\Sigma_1^{-1} + \Delta_1)(\mathbf{Z}_1 + \mathbf{R}_1)^\top \mathbf{Y}$$
$$= \hat{\theta}_c^* + (\Delta_1 \hat{\mathbf{Z}}_1^\top + \Sigma_1^{-1}\mathbf{R}_1^\top)\mathbf{Y}$$

where $\Delta_1 = -\Sigma_1^{-1}\Omega_1(\Sigma_1 + \Omega_1)^{-1}$, $\hat{\theta}_c^*$ is the OLS estimate of $\theta$ for the true $\Phi$ and the second line follows from the Woodbury matrix identity. The norm of $\Delta_1$ is related to the condition number of $\Sigma_1$. The expectation of the first term is $\theta$, and the expectation of the remaining terms is the bias.

Now, we can do the same thing for the privileged estimator. Let's start with $T = 2$.

$$
\begin{aligned}
\hat{\theta}_p &= \hat{\Sigma}_1^{-1}\hat{\mathbf{Z}}_1^\top\hat{\mathbf{Z}}_2\hat{\Sigma}_2^{-1}\hat{\mathbf{Z}}_2^\top Y \\
&= (\Sigma_1^{-1} + \Delta_1)(\mathbf{Z}_1 + \mathbf{R}_1)^\top(\mathbf{Z}_2 + \mathbf{R}_2)(\Sigma_2^{-1} + \Delta_2)(\mathbf{Z}_2 + \mathbf{R}_2)^\top\mathbf{Y} \\
&= \hat{\theta}_p^* + \hat{A}_1(\Sigma_2^{-1}\mathbf{R}_2^\top + \Delta_2\hat{\mathbf{Z}}_2^\top)\mathbf{Y} + (\Sigma_1^{-1}\hat{\mathbf{Z}}_1^\top\mathbf{R}_2 + \Sigma_1^{-1}\mathbf{R}_1^\top\hat{\mathbf{Z}}_2 + \Delta_1\hat{\mathbf{Z}}_1^\top\hat{\mathbf{Z}}_2)\hat{\beta} \, .
\end{aligned}
$$

Thus, the difference in bias between the two estimators is

$$
\begin{aligned}
\mathbb{E}[\hat{\theta}_c - \hat{\theta}_p] = &\underbrace{\mathbb{E}[\hat{\theta}_c^* - \hat{\theta}_p^*]}_{=0} \\
&+ \mathbb{E}[(\Delta_1\hat{\mathbf{Z}}_1^\top + \hat{\Sigma}_1^{-1}\mathbf{R}_1^\top)\mathbf{Y} - \hat{A}_1(\Delta_2\hat{\mathbf{Z}}_2^\top + \hat{\Sigma}_2^{-1}\mathbf{R}_2^\top)\mathbf{Y} \\
&- (\Sigma_1^{-1}\hat{\mathbf{Z}}_1^\top\mathbf{R}_2 + \Sigma_1^{-1}\mathbf{R}_1^\top\hat{\mathbf{Z}}_2 + \Delta_1\hat{\mathbf{Z}}_1^\top\hat{\mathbf{Z}}_2)\hat{\beta}]
\end{aligned}
$$

More generally, we can express this difference recursively as below.

**Proposition 3.** *Let $\hat{\theta}_p := \hat{A}_1 \cdots \hat{A}_T\hat{\beta}$ be a privileged estimator using a linearly biased representation $\hat{\Phi}$, and let $\hat{A}_t^*$ be the same estimator using an unbiased representation $\Phi^*$. Then, the bias of $\hat{\theta}_p$ is*

$$
\mathbb{E}[\hat{\theta}_p - \theta] = \mathbb{E}[\hat{\theta}_p - \hat{\theta}_p^*] = \mathbb{E}[E_T\hat{\beta}^* + (\hat{A}_1 \cdots \hat{A}_T)(\hat{\beta} - \hat{\beta}^*)] \, ,
$$

*where $E_t$ is the compounded error in transition dynamics, computed recursively as follows*

$$
E_t := (\hat{A}_1 \cdots \hat{A}_t) - (\hat{A}_1^* \cdots \hat{A}_t^*) = E_{t-1}\hat{A}_t^* + (\hat{A}_1 \cdots \hat{A}_{t-1})(\hat{A}_t - \hat{A}_t^*) \, ,
$$

*with $E_0 = 0$. In the worst case, the bias of $\hat{\theta}_p$ grows exponentially with $T$.*

## D  Privileged time series representation learners

We expand on the description of the greedy representation learner (GRL) described as a special case of CRL in Section 3.3. To avoid the information loss of SRL, we consider its conceptual opposite, using privileged time series information *only* to predict the outcome. To do this, a linear output layer $\hat{\theta}_t^\top\hat{Z}_t$ is used to predict $Y$ at every time step $t$. Recall that, by Assumption 1, the expected outcome is linear in the latent state at *any* time step. The method is related to multi-view learning, in which prediction of the same quantity is made from multiple "views" (Zhao et al., 2017). We dub the model *greedy privileged representation learner* (GRL), which minimizes the objective

$$
\mathcal{L}_{\text{GRL}}(\hat{\Phi}, \{\hat{\theta}_t\}) := \frac{1}{NT}\sum_{i=1}^N\sum_{t=1}^T w_t\|\hat{\theta}_t^\top\hat{\Phi}(x_{i,t}) - y_i\|_2^2. \tag{9}
$$

During inference this algorithm returns $\hat{h}_{\text{P}}(\cdot) = \hat{\theta}_1^\top\hat{\Phi}(\cdot)$. Compared to the objective of CRL in 6, we introduce an additional hyperparameter $\lambda \in (0, 1)$ to place more weight on the loss term that is relevant at inference time. As a consequence, we choose $w_t = \lambda$ for $t = 1$ and $w_t = 1 - \lambda$ otherwise. We expect GRL to have less bias than SRL, but higher variance since less structure is imposed on the representation $\hat{\Phi}$.

## E  Experiment setup & data processing

### E.1  Detailed experiment setup

In the following we give a detailed description of the experimental setup used to obtain the results presented in Section 4 as well as the additional results that are part of this section. For a given data set, we select a combination of training set sizes and sequence length. For each unique combination of these parameters the models of interest are trained repeatedly with different random sampling. For each repetition the data is split into a train and a test set randomly before hyperparameter tuning and model training are performed. At last each model's predictions on the test set are scored by computing the coefficient of determination $R^2$. On synthetic data the test set contains 1000 samples. In the case of real-world data, where samples are limited, we test on 20% of all available data.

| Hyperparameter | Description | Used in Algorithm | Value Range |
|---|---|---|---|
| $n_{RF}$ | number of random features | all random feature methods | $[0.05m, 0.8m]$ |
| $\gamma_{RRF}$ | bandwith parameter | Random ReLU methods | $[0.01, 10]$ |
| $\gamma_{RFF}$ | bandwith parameter | Random Fourier methods | $[0.001, 0.1]$ |
| $\lambda$ | loss function parameter | GRL & CRL | $[0, 1]$ |

Table 1: Overview of all hyperparameter determined by hyperparameter tuning. $m$ denotes the number of samples, meaning that $n_{RF}$ is chosen from different ranges depending on the size of the training set.

The preprocessing used for real-world data and the generation procedure of synthetic data is unique to each data set. We refer to the data set specific subsections for detailed descriptions of how each data set is processed. During the experiments each model uses standardized data for training and inference. To perform the data rescaling we use the StandardScaler implementation that is part of scikit-learn (Pedregosa et al., 2011).

**Hyperparameter tuning.** The tuning of hyperparameters is carried out for each repetition and is implemented using random search and five-fold cross-validation. Each hyperparameter is sampled from a fixed interval of possible values. An overview of the ranges of different hyperparameters determined through random-search is provided in Table 1. For the experiments with variants of generalized LuPTS we sample ten sets of hyperparameters before retraining on all training data using the best set of parameters. For the representation learners we merely sample five values for $\lambda$.

**Neural network training.** The training of neural networks involves many choices and hyperparameters. We choose PyTorch's implementation (Paszke et al., 2019) of the Adam optimizer (Kingma and Ba, 2017) to train the representation learning models. If not specified otherwise the results shown in this project are obtained using a learning rate of $0.0001$, a batch size of $30$, leaky ReLU activations and a maximum of $1500$ training epochs. In the case of neural network models, the sample sizes reported as part of the experiments denote the combined size of the training and validation set, where the validation set contains 20% of those samples. We use early stopping during the training process by keeping track of the validation loss. If a model does not improve the validation loss over a waiting period of 200 epochs we stop training early and set the network parameters to the values that obtained the lowest validation loss up until that point. In order to make sure that results are not dependent on the specific choice of the parameters just described, we performed additional experiments with different parameter choices. The results were found to be robust to small changes in these parameters.

**Generalized distillation.** In order to compare our algorithms to the alternative of using generalized distillation for privileged time series as presented by Hayashi et al. (2019), we implemented a model that (i) produces hypotheses of the same class as our other algorithms and (ii) that adopts the learning paradigm of a student model incorporating soft targets produced by a teacher model into its loss function. For tabular data our teacher model is a multi-layer-perceptron (MLP) with $T * k$ input neurons such that all $\{X_t\}$ are concatenated and then used as input. The teacher MLP makes use of five hidden layers, each consisting of 100 neurons. In the case of the image data generated by **Clocks-LGS**, the teacher uses an implementation of LeNet-5 on all variables $X_t$ (while sharing the encoder parameters) before concatenating the 25-dimensional output of this encoder from different time steps. This combined representation is then processed by an MLP with a single hidden layer with 25 neurons. The loss function used for the student model producing the estimate $\hat{h}_{\mathrm{c}}$ is architecturally identical to the classic representation learner used in each of the experiments. When training the student model the mean squared error on the data and the error corresponding to the soft targets of the teacher model are combined via a hyperparameter $\lambda$:

$$\mathcal{L}_{GD} := \lambda \frac{1}{m} \sum_{i=1}^{m} \|\hat{h}_{\mathrm{c}}(x_i) - y_i\|_2^2 + (1 - \lambda) \frac{1}{m} \sum_{i=1}^{m} \|\hat{h}_{\mathrm{c}}(x_i) - \hat{h}_{\mathrm{teacher}}(x_i)\|_2^2$$

The hyperparameter is determined via hyperparameter tuning in all repetitions as described for other hyperparameters in this section. All other training procedures follow the same logic as described above.

**Resources.** For the training of the representation learning algorithms we use a cluster of graphics processing units (GPUs) in order to reach the number of experiment repetitions required for our work. A single experiment like shown in Figure 5c takes several hours on 100 NVIDIA Tesla T4 GPUs. While the random feature methods do not require GPU training, they still require hyperparameter tuning which is why we compute results such as presented in Figure 4 on many CPU cores in parallel. While the experiments on neural networks cannot reasonably be reproduced on a single desktop machine, this is still possible within a few days for the random feature methods.

**Latent variable recovery and SVCCA.** In order to assess to what extent a representation has been found that is linearly related to the true latent variables we use SVCCA as described by Raghu et al. (2017), meaning we first use PCA retaining at least 99% of variation before then applying CCA. For the visualization shown in Figure 7b we construct a grid of points ($150 \times 150$) around the origin and assign each point a unique color. Then we compute an image using the observation generating function $\Psi : \mathcal{Z} \to \mathcal{X}$ of the **Clocks-LGS** data set for each point. The observations are passed to the encoder $\hat{\Phi}$ of the two representation learners producing estimates of the latent variables. We then map the estimates to the ground truth linearly using SVCCA before plotting the result.

### E.2 Alzheimer progression

To test our algorithms on the task of predicting the progression of Alzheimer's disease (AD) we use an anonymized data set obtained through the Alzheimer's Disease Neuroimaging Initiative (**ADNI**) (ADNI, 2004) under the LONI Research License. The initiative is large multi-site research study on the brains of over 2000 AD patients which collects many features such as genetic, imaging and biospecimen biomarkers. The data consists of measurements taken every 3 months with some observations missing. The outcome of interest in our experiments is the Mini Mental State Experiment (MMSE) score 48 months after the first measurement(Galea and Woodward, 2005). Privileged information are the measurements taken between those time points, at 12, 24 and 36 months into the program.

**Data processing.** The processing procedure used in this project is borrowed directly from the work of Karlsson et al. (2022). There is a large amount of missing information in the ADNI data set. The missingness varies with the time of when measurements were taken. Further some subjects were not present at some of the follow-up examinations. To deal with the missingness patients without an observation for the final follow up (the outcome $Y$) are excluded from our experiments. Further, we also require that patients are present at all intermediate time steps (12, 24 and 36 months after the first measurement) which we use as privileged information. We one-hot encode categorical features and exclude features for which more than 70% of the observations are missing. To deal with the remaining missing values, mean imputation is used. Due to the filtering that we apply as a result of the missing data we only obtain 502 suitable sequences that we can use for our experiments.

### E.3 Traffic data

The **Traffic** data set (Hogue, 2012) obtained through the UCI machine learning repository (Dua and Graff, 2017) contains hourly measurements of the traffic volume as well as weather features and holiday information. The raw data contains 48.204 records. An overview of all available features is given in Table 2.

**Data processing.** We noticed extreme outliers in the data set as well as implausible numerical values for the temperature and rain features. Further, records for some of the hours of the timeframe (2012 - 2018) covered by this data set are missing. To deal with the extreme outliers we calculate the mean and standard deviation of each feature and remove records which contain values that are further than six standard deviations from the mean of a particular feature. We also remove a feature entirely if there is no variation left after this filtering. This is the case for the snowfall feature as snow is very rare in Minneapolis. From the date and time of each record we calculate the weekday which we add

| Feature | Type | Description |
|---|---|---|
| Date Time | Timestamp | date and time (CST) |
| Holiday | String | name of holiday if applicable |
| Weather Description | String | brief free text description of the weather |
| Weather Main | Categorial | contains categories like clear, clouds, or rain |
| Rain_1h | Numerical | rain in $\frac{L}{h\,m^2}$ |
| Snow_1h | Numerical | snow in $\frac{L}{h\,m^2}$ |
| Temp | Numerical | temperature in Kelvin |
| Traffic Volume | Numerical | hourly reported westbound traffic volume |

Table 2: Features available in the **Traffic** data set.

as a one-hot encoded feature and also represent the hour of the day $h \in \{0, 1, \ldots 23\}$ as two separate periodic features given by

$$t_{\text{periodic}} = \left[ \sin\left(\frac{2\pi \cdot h}{24}\right),\ \cos\left(\frac{2\pi \cdot h}{24}\right) \right]. \tag{10}$$

This ensures that a timestamp just before midnight produces similar features compared to just after midnight. We one-hot encode the holiday information, making no difference between different types of holidays, and make this feature persist over a full calendar day. In the original data set the holiday information is only specified for the first hour of the day. The column Weather_Main contains some weather conditions that are very rare, such as smoke and squall. As a consequence we group the different conditions before encoding them as binary variables. In particular we make drizzle, rain and squall one single feature while also grouping together fog, haze, mist and smoke as they all affect visibility.

**Time series selection.**  After this preprocessing, that leaves only numerical values and one-hot encoded categorical values, we group the data together as time series used for the experiments. In order to do so we specify a desired sequence length $T + 1$ and a sequence step size in hours. With this information we iterate through the data set assembling time series with i) no values missing ii) the correct length and step size and iii) at least a seven hour gap between each pair of sequences. The third condition is introduced to make sure one does not end up with very similar cases (for short sequences in particular) in training and test set.

### E.4  Square-Sign

The **Square-Sign** data set serves as a test environment for learning from privileged time series information where one can assure the conditions necessary for Assumption 1 to hold. In particular this means creating a linear-Gaussian system which remains unobserved and combining it with an observation generating function $\psi : \mathcal{Z} \to \mathcal{X}$.

**Latent linear-Gaussian system.**  The first component that makes up the generation process for **Square-Sign** (and **Clocks-LGS**) is the linear-Gaussian system which is latent, just as depicted on the right side of Figure 1b with $Z_t \in \mathbb{R}^d$. The first step in the data creation process is sampling each of the $d$ components of $Z_0$ from $\mathcal{N}(0, 5)$. Then the subsequent latent variables $Z_{t+1}$ are computed as

$$Z_{t+1} := A_t^\top Z_t + \epsilon,\ \epsilon \in \mathbb{R}^d,\ \epsilon_j\, \mathcal{N}(0, 1) \,.$$

For the outcome we use the same form but with different dimensionality:

$$Y := \beta^\top Z_T + \epsilon_y,\ \epsilon_y \in \mathbb{R}^q$$

Off-diagonal elements of the transition matrices $A_t \in \mathbb{R}^{d \times d}$ are sampled from a Normal distribution $\mathcal{N}(0, 0.2)$ while the diagonal elements are set to one. In a second step we compute the spectral radius of the randomly created matrices $A_t$ via eigenvalue decomposition, obtaining the components $U \Lambda U^\top$. We then set the spectral radius to a predefined value $\lambda_{max} = 1.3$ and reassemble the matrix as

$$A_t \leftarrow U \frac{\lambda_{max}}{\lambda_s} \Lambda U^\top \,.$$

The coefficients of $\beta$ are drawn from the same normal distribution as the ones of $A_t$ but undergo no further changes.

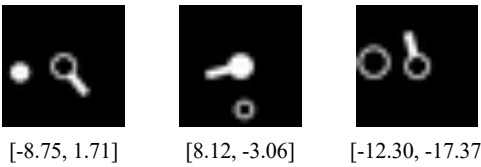

| [-8.75, 1.71] | [8.12, -3.06] | [-12.30, -17.37] |

Figure 8: Pairs of corresponding latent vectors $Z_t$ and images as generated in **Clocks-LGS**.

**Observation generating function.** As the dimensionality $d$ of the latent space $\mathcal{Z}$ is not fixed we use an observation generating function that is not restricted to a specific value of $d$. For each element in $Z_t \in \mathcal{Z} = \mathbb{R}^d$ we create two elements in $X_t \in \mathcal{X} = \mathbb{R}^{2d}$ by denoting its sign separately from its square. This gives the following nonlinear observation generating function:

$$X_t \coloneqq \psi(Z_{(t)}) = [Z_{(t,1)}^2, \operatorname{sgn}(Z_{(t,1)}), \ldots, Z_{(t,d)}^2, \operatorname{sgn}(Z_{(t,d)})]^\top$$

### E.5 Clocks-LGS

This data set serves the purpose of testing our algorithms on a different modality with high dimensional data. In particular the idea was to use image data as this is a domain where neural networks have been very successful. For this reason we combine a latent dynamical system with an image generation process which we explain in detail in this section.

**Latent linear-Gaussian system.** We use exactly the same setup as we do for the **Square-Sign** latent dynamical system as described in Section E.4. The only difference here is the dimensionality of the latent variables, transition matrices and the outcome. For **Clocks-LGS** we generally have $d = 2$ and $q \in \{1, 2\}$.

**Image generation.** The second part of **Clocks-LGS** is creating images from two dimensional latent vectors $Z_t = [Z_t^{(1)}, Z_t^{(2)}]^\top$. The goal was to keep it the process simple while using small black and white images of 28×28 pixels. In addition we wanted each image to have no ambiguity with respect to the latent state it represents. We represent the first component by a clock hand mounted at the image center. One can think of $Z_t^{(1)}$ as the angle in radian, meaning the hand points straight up for $Z_t^{(1)} = 0$ or $Z_t^{(1)} = 2\pi$ and straight down for $Z_t^{(1)} = \pi$. To visualize a full rotation we increase the size of the cirle around the image center in discrete steps for each mutliple of $2\pi$. For negative values the circle is empty (black) while it is filled (white) for positive values. For the second component we make use of the same logic but instead of a clock hand, we only use a circle that orbits the image center. The two hands cannot obscure each other as the orbiting cirle uses a larger radius. Figure 8 shows three examples of pairs of corresponding latent vectors and generated images.

### E.6 PM$_{2.5}$ air quality

Due to health concerns the air quality in Chinese cities has become an important topic. The **PM$_{2.5}$** data set contains hourly meteorologic information and the concentration of small particles (PM$_{2.5}$) for the cities Beijing, Shanghai, Guangzhou, Chengdu and Shenyang (Liang et al., 2016). The individual features available for all cities are listed in Table 3. In addition to the features listed, the data includes the date and time of each record. Just like in the preprocessing of **Traffic** we compute a periodic time feature using expression 10 to represent the time of day of each record. For each numerical feature we calculate the mean and standard deviation and remove rows with values that are more extreme than six standard deviations from the mean. We also remove rows with missing categorical features, which are then represented as one-hot vectors. Apart from differences in the preprocessing we consider the same prediction task as Karlsson et al. (2022) which is predicting the future particle concentration for a fixed time horizon given current observations.

| Feature | Type | Description |
|---|---|---|
| season | Numerical | season (1 to 4) of the data in this row |
| PM | Numerical | $PM_{2.5}$ particle concentration in $\mu g/m^3$ |
| DEWP | Numerical | dew point in $°C$ |
| TEMP | Numerical | air temperature in $°C$ |
| HUMI | Numerical | humidity in % |
| PRES | Numerical | atmospheric pressure in hPa |
| cbwd | Categorical | combined wind direction in {N,W,S,E, NW, SW, NE, SE} |
| Iws | Numerical | cumulated wind speed in $m/s$ |
| precipitation | Numerical | hourly precipitation in $mm$ |
| Iprec | Numerical | cumulated precipitation in $mm$ |

Table 3: Features available as part of the $\mathbf{PM_{2.5}}$ data set on the air quality of five large Chinese cities.

# F   Additional experiment results

In the course of this section we present a larger scope of our experimental results. We demonstrate the predictive accuracy of the algorithms introduced in Sections 3.2 and 3.3 in terms of the mean coefficient of determination $R^2$ over different settings on five data sets. The variation of the results over repetitions is represented by the shaded areas in the visualizations, which denotes one standard deviation above and below the mean value. We also show empirically that the bias of generalized LuPTS increases with the number of privileged time steps when a poor representation function is used. This can be seen in Figure 16a where we test privileged and classical learners over 500 different systems of the **Square-Sign** type over different sequence lengths. In addition we further illustrate how privileged information can improve the recovery of latent variables of the data generating process by providing more visualizations in the style of Figure 7a on the **Clocks-LGS** and **Square-Sign** data set. These can be found in Figures 13 and 16.

In the following we first provide the experiment details for visualizing the phase transition of generalized LuPTS as seen in Figure 2. In the subsections thereafter the material is organized by data set.

## F.1   Two regimes of generalized LuPTS

As seen in Figure 2 and demonstrated by Proposition 1, generalized LuPTS becomes equivalent to the corresponding classical learner when the number of features $\hat{d}$ is larger than the number of samples $m$. In the following we provide the experiment details that led to Figure 2.

In order to evaluate the dependency on the number of features we used linear LuPTS and OLS on a synthetically generated linear-Gaussian system as displayed in Figure 1a. We use the same setup as for the latent dynamics in **Square-Sign** but without a nonlinear observation generating function. For each number of features $k = \hat{d} = d$ we sample 50 such systems with different dynamics, each producing a training set with $m = 100$ and test set of 1000 samples. The systems are all configured with $T = 3$ and $q = 10$. We train and score both estimators on all of the data generating systems before computing the mean coefficient of determination over all systems with the same number of features.

## F.2 Alzheimer progression

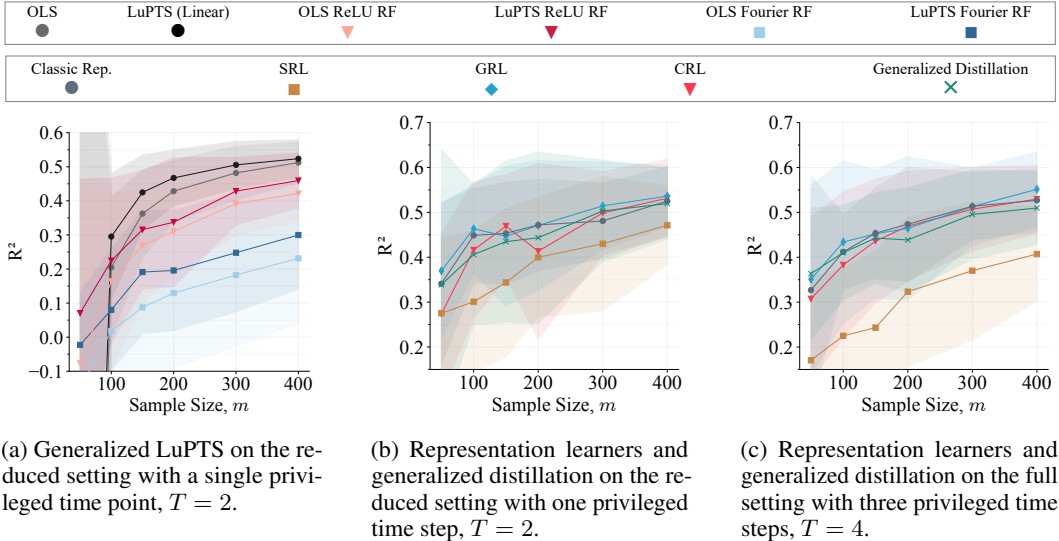

(a) Generalized LuPTS on the reduced setting with a single privileged time point, $T = 2$.

(b) Representation learners and generalized distillation on the reduced setting with one privileged time step, $T = 2$.

(c) Representation learners and generalized distillation on the full setting with three privileged time steps, $T = 4$.

Figure 9: Generalized LuPTS and the representation learning algorithms tested in terms of their predictive accuracy on different settings of the **ADNI** prediction task. Each experiments are based on 25 repetitions while the random feature experiment consists of 60 repetitions.

## F.3 Traffic data

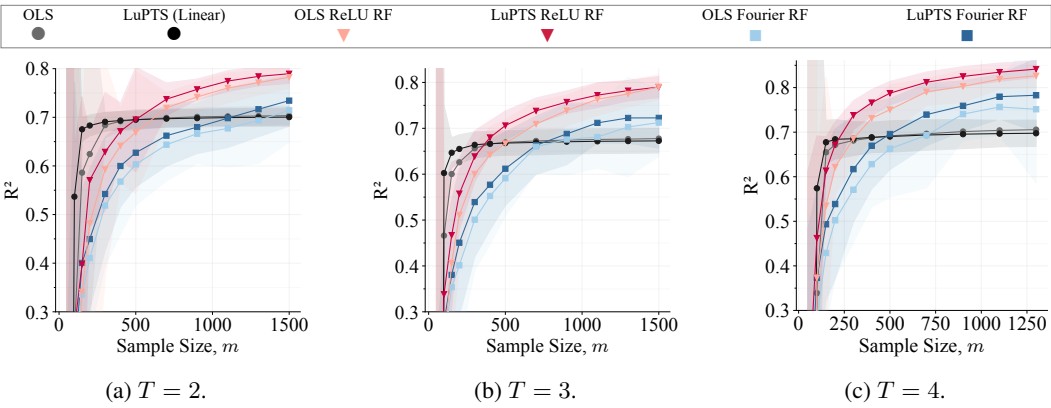

(a) $T = 2$.

(b) $T = 3$.

(c) $T = 4$.

Figure 10: Prediction accuracy of the different variants of generalized LuPTS on **Traffic** with varying sequence lengths. Based on 60 repetitions and four hour steps in each time series.

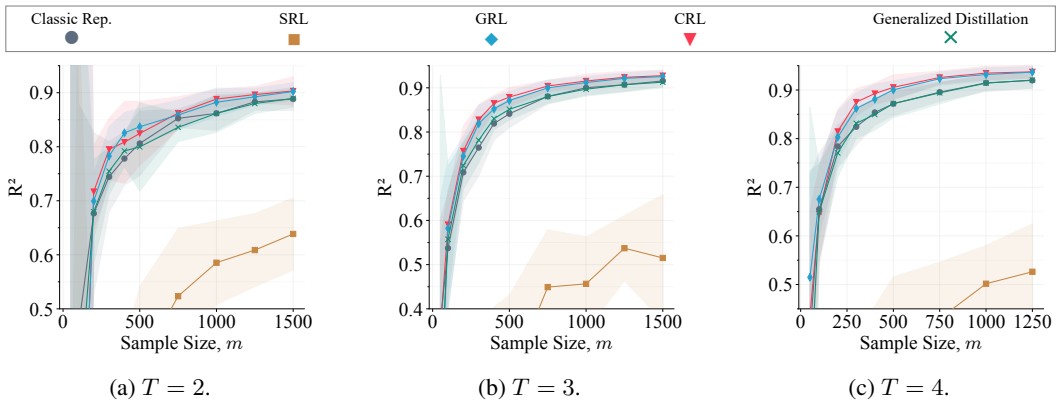

Figure 11: Prediction accuracy of the different representation learning algorithms and generalized distillation on **Traffic** with varying sequence lengths. The results are based on 25 repetitions and four hour steps in each time series.

## F.4 Clock_LGS

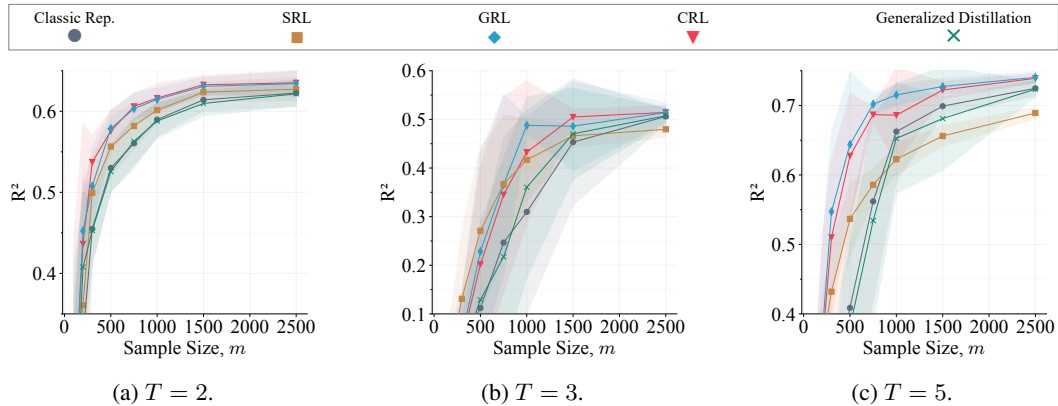

Figure 12: Prediction accuracy of the representation learning algorithms and generalized distillation on **Clocks-LGS** with a two dimensional outcome $q = 2$ and varying sequence lengths, based on 25 repetitions.

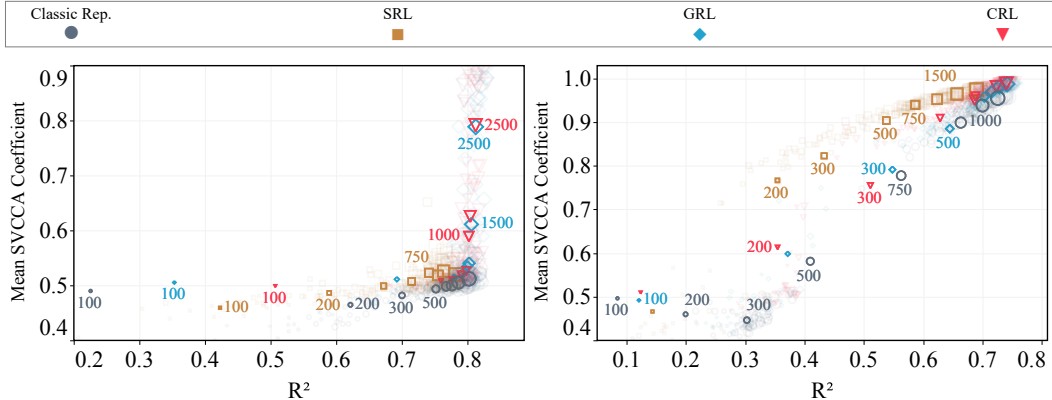

Figure 13: Mean SVCCA coefficients and $R^2$ for the representation learners on the **Clocks-LGS** prediction task. The experiment was set up for sequences of length six ($T = 5$), with outcomes of different dimensionality: $q = 1$ on the left and $q = 2$ on the right. Solid marks represent the mean over 25 repetitions, while the faded marks denote the individual training runs. The annotations refer to the size of the training sets.

## F.5 Square-Sign

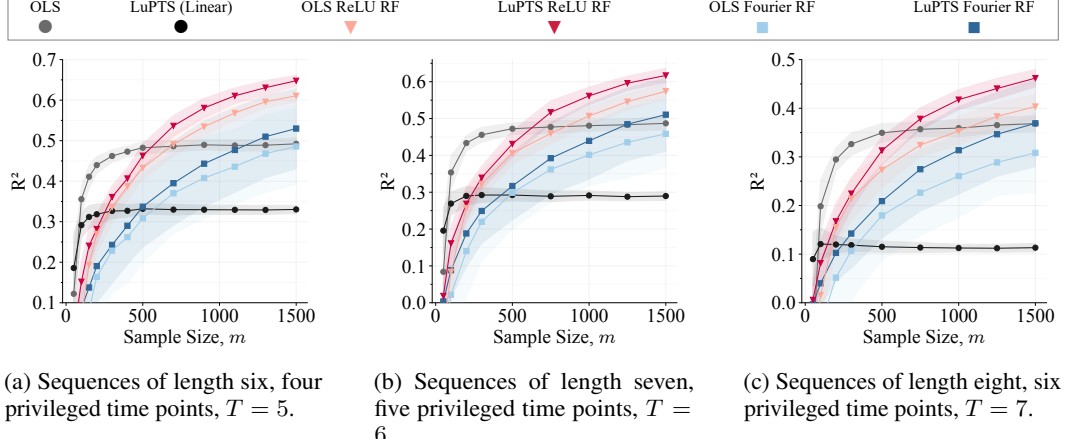

(a) Sequences of length six, four privileged time points, $T = 5$.

(b) Sequences of length seven, five privileged time points, $T = 6$.

(c) Sequences of length eight, six privileged time points, $T = 7$.

Figure 14: Predictive accuracy ($R^2$) of the different variants of generalized LuPTS applied to the prediction task offered by **Square-Sign**. The DGP was configured with different sequence lengths and the experiments represent 60 repetitions.

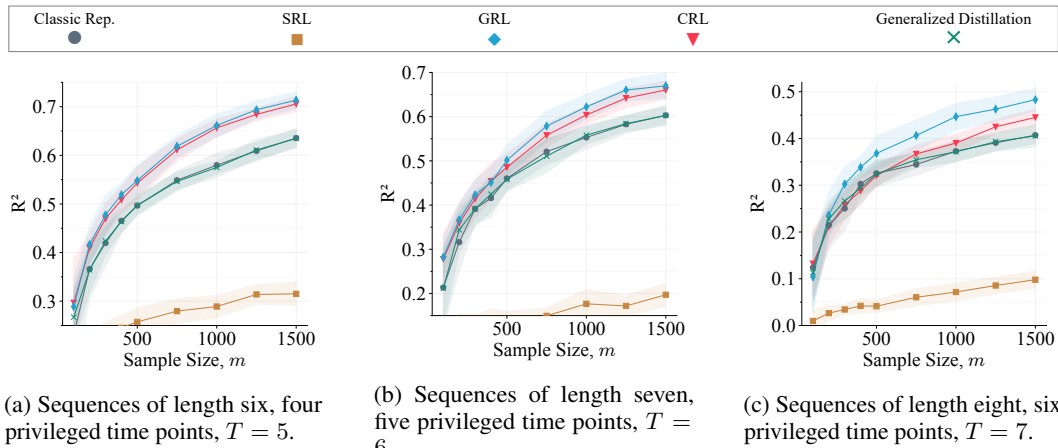

(a) Sequences of length six, four privileged time points, $T = 5$.

(b) Sequences of length seven, five privileged time points, $T = 6$.

(c) Sequences of length eight, six privileged time points, $T = 7$.

Figure 15: Predictive accuracy of the different representation learners introduced in Section 3.3 when applied to the **Square-Sign** data set for different sequence lengths. The experiments are based on 25 repetitions.

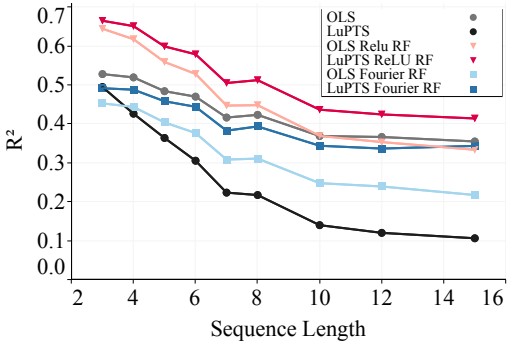
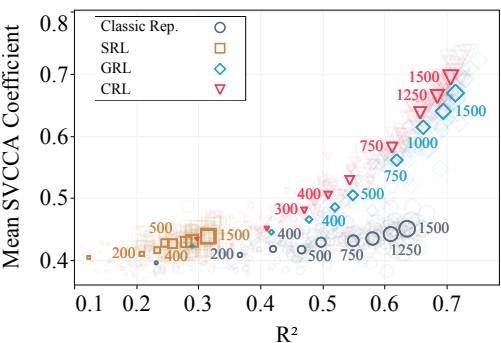

(a) Mean $R^2$ over 500 repetitions for different sequence lengths on **Square-Sign** ($d = 10$, $q = 3$) with a sample size of $m = 1000$. Each run is performed on a different set of randomly sampled transition dynamics.

(b) Mean SVCCA coefficients and $R^2$ for the representation learners on the **Square-Sign** data set configured with ($T = 5$, $d = 10$, $q = 3$). Solid marks represent the mean over 25 repetitions for a fixed training set size while the faded marks denote individual runs. The annotations refer to the number of samples in the training data.

Figure 16: Generalized LuPTS applied on the **Square-Sign** task for different sequence lengths (left panel), reporting the mean coefficient of determination $R^2$ over different systems. The right panel shows analysis of the predictive accuracy and latent recovery of the representation learners in the style of Figure 7a.

## F.6 PM$_{2.5}$ air quality

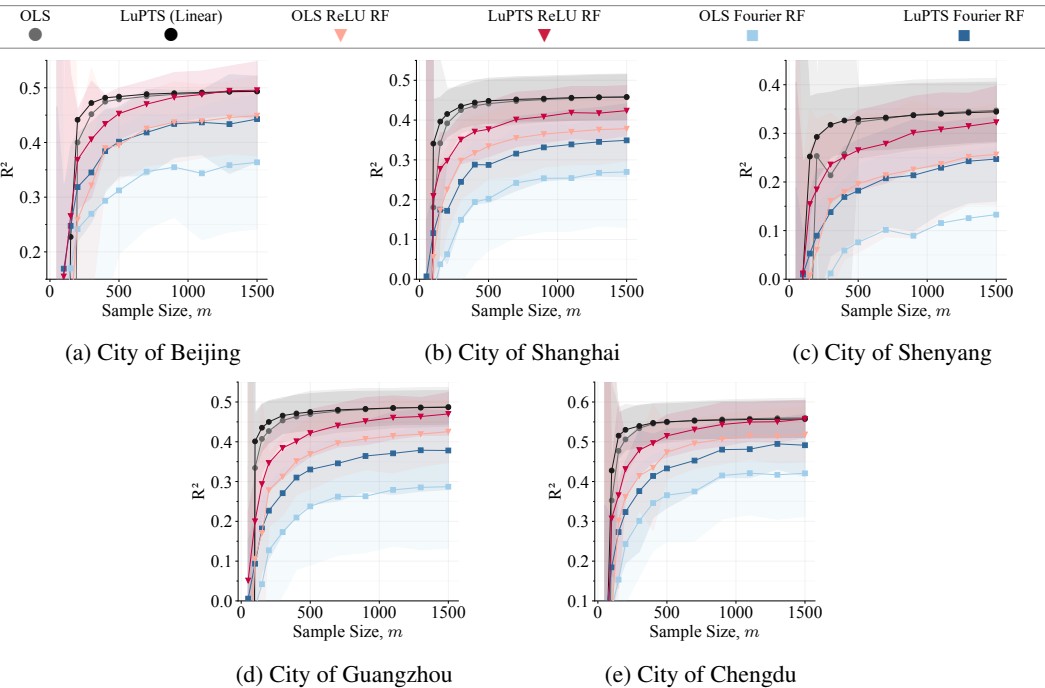

(a) City of Beijing

(b) City of Shanghai

(c) City of Shenyang

(d) City of Guangzhou

(e) City of Chengdu

Figure 17: Predictive accuracy of the different generalized LuPTS variants on the prediction task posed by the **PM$_{2.5}$** data set. All experiments use time series of length five ($T = 4$), where each time step is two hours long. The results were computed based on 60 repetitions.

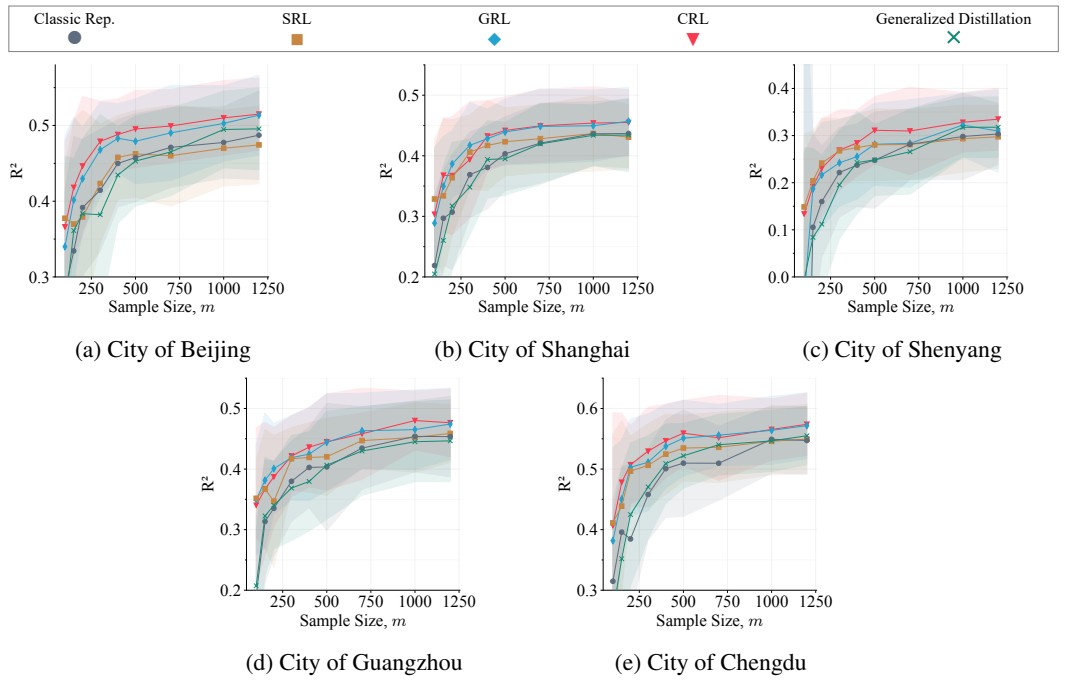

(a) City of Beijing     (b) City of Shanghai     (c) City of Shenyang

(d) City of Guangzhou     (e) City of Chengdu

Figure 18: Evaluation of the sample efficiency of the represenation learning algorithms of Section 3.3 on the $PM_{2.5}$ air quality prediction task. All experiments use time series of length five ($T = 4$), where each time step represents two hours. The results were computed based on 25 repetitions.