# OpenReview forum: "Efficient learning of nonlinear prediction models with time-series privileged information"
_NeurIPS.cc/2022/Conference — NeurIPS 2022 Accept_

### Official Review · Reviewer_LQNq · 2022-07-08

**Rating:** 5
**Confidence:** 4
**Soundness:** 3 good
**Presentation:** 3 good
**Contribution:** 3 good

**Summary:**

The manuscript proposes to predict nonlinear outputs using time series privileged information during training. The privileged information is not used for inference. The authors prove in Theorem 1 that under some conditions, "the generalized LuPTS is never worse in expectation than the classical learner". Experiments on both synthetic and real-world data sets show that using privileged information increases the sample efficiency and helps latent variable recovery.

**Questions:**

* Stochastic process

If I understand correctly, the latent states $Z_t$ are assumed to be from a stochastic process, is it?
It is yet not clear from the main paper whether $Z_t$ is stationary or not.
I see some conditions of matrix $A$ in Appendix (line 659), but I did not find the constraints of generating $A$. That is, it is not clear why $A$ is designed to be like that.

* OLS Linear estimator

What is the difference between OLS (line 100) and the linear regression?
$\hat{\theta}_C$ in eq(3) seems to be exactly the same as linear regression.

* $R^2$

What is $R^2$ in Section 4?

* Efficiency

The manuscript claims in line 237 that the main goal is to improve learning efficiency.
How "learning efficiency" is defined and estimated in the manuscript?





**Limitations:**

The authors addressed the limitations.

**Strengths And Weaknesses:**

* Originality

From the title, it is not clear how much this manuscript differs from [1] Karlsson et al. (2021) (https://proceedings.mlr.press/v151/k-a-karlsson22a/k-a-karlsson22a.pdf)
While in the main text, the manuscript tackles the task by modeling with latent variables for each observation, which is a major difference.
If I understand correctly, Theorem 1 in the manuscript is similar to Theorem 1 in [1] with some changes of conditions.
The manuscript should better formulate the difference from [1]

* Quality

The manuscript is technically sound.

* Clarity

The manuscript is overall clearly written. Some (minor) suggestions are as follows.

1. "**Unobserved**" applies to Figure 1 (b) but not Figure 1 (a).
2. It is not clear what random variables the "distribution $p$" models in line 62.
3. "the matrix inverse $(\cdot)^{-1}$" in line 104 can not be found in the text.
4. Figure 3 (b) is not informative enough.
5. It is not clear what the metric in experimental evaluation is used. What is $R^2$ in Section 4?


* Significance

The algorithm proposed in the manuscript would benefit the time series modeling community

---

[1] Karlsson, Rickard KA, et al. "Using time-series privileged information for provably efficient learning of prediction models." International Conference on Artificial Intelligence and Statistics. PMLR, 2022.

---

> ### Author Response · Authors · 2022-07-31
> **Response to review by Reviewer LQNq**
>
> We thank LQNq for their insightful feedback!
>
> **From the title, it is not clear how much this manuscript differs from Karlsson et al. (2021)**
>
> The key word in the title is *nonlinear*. The main contribution of our work is the generalization to prediction of nonlinear outcomes—this is not supported by Karlsson et al. In the original submission, we point out key differences on l.32–38, l.84–86, l.121–127.
>
> Differences in theory are captured mainly in Theorem 1. The reviewer is correct that the main difference between theorems is that our result has different (less strict) conditions: the observed system is no longer assumed to be a linear-Gaussian Markov chain. As a result, the goal of learning, the mapping $E[Y \mid X_1]$ from input to outcome can be nonlinear. Instead, we assume that observations can be mapped to such a chain using a (nonlinear) representation known up to linear transform. In Section 3.2 and Appendix B, we show that such a representation can be learned consistently using random feature maps.
>
> To predict nonlinear outcomes, we also propose new learning algorithms based on kernels, random feature maps and representation learning using neural networks. Neither was previously used by Karlsson et al. The methods are then evaluated in a new set of experiments.
>
> **Are the latent states assumed to be from a stochastic process? Is $A$ stationary? I see some conditions of matrix A in Appendix, but I did not find the constraints of generating $A$. Why is $A$ designed to be like that.**
>
> Yes, this is correct. The latent states form a stochastic process, with finite horizon T, assumed for Theorem 1 to be linear-Gaussian, see the definition of $Z_t$ in Assumption 1. The process is not assumed to be stationary, i.e., $A_t$ and $\sigma_t$ can vary with $t$. We have updated the notation on l.82 slightly to clarify that $A_1, ..., A_{T-1}$ are (potentially) different matrices.
>
> The choice of $A$ is not constrained in general. $A$ was chosen for synthetic experiments in order to create an interesting learning task (a similar procedure was used by Karlsson et al (2022)). It is not required by our theoretical results and we expect many choices of $A$ to produce similar empirical results. In the experiments on real-world datasets, we have no control over the data generating process.
>
> **What is the difference between OLS and the linear regression?**
>
> They are the same. OLS (ordinary least squares) is the most common method for fitting a linear regression i.e., by minimizing the empirical average least squares error. The name OLS is well established in machine learning and statistics, see e.g., its uncommented use in [1].
>
> **What is R^2 in Section 4?**
>
> $R^2$ is the “coefficient of determination”, as stated on l.200 in the original submission. It is one of the most commonly used metrics to evaluate regression models in statistics and machine learning. For this reason, we refrain from stating a mathematical definition. See, for example, https://en.wikipedia.org/wiki/Coefficient_of_determination (first definition of $R^2$) and https://scikit-learn.org/stable/modules/generated/sklearn.metrics.r2_score.html, which is the implementation we used in our experiments.
>
> **The manuscript claims that the main goal is to improve learning efficiency. How "learning efficiency" is defined and estimated in the manuscript?**
>
> Sample efficiency classically refers to the ability of a statistical estimator to achieve smaller error/risk than other estimators with the same number of samples [2, Chapter 20]. For unbiased estimators, this coincides with their variance. For learning algorithms, efficiency is typically measured by how the expected error of a learning algorithm varies with the number of training samples. This is the notion we adopt here and it can be precisely defined; a learner is more efficient if the expected risk $\bar{R}_D$ is smaller for the same number of samples $m = |D|$. Theorem 1 shows that learning using privileged information is more efficient than non-privileged learning with empirical risk minimization in the same hypothesis class under appropriate conditions. We have clarified the connection between risk and sample efficiency on l.65–72 in the revision. Empirically, we estimate efficiency by comparing the prediction error for different estimators as we vary the number of training samples.
>
> **Minor suggestions**
>
> We thank reviewer LQNq for these suggestions and will incorporate them in the final version of the paper.
>
> **References**
>
> [1] Hastie, T., Tibshirani, R., Friedman, J. H., & Friedman, J. H. (2009). The elements of statistical learning: data mining, inference, and prediction (Vol. 2, pp. 1-758). New York: springer.
>
> [2] Dekking, F. M., Kraaikamp, C., Lopuhaä, H. P., & Meester, L. E. (2005). A Modern Introduction to Probability and Statistics: Understanding why and how (Vol. 488). London: Springer.

---

> > ### Comment · Reviewer_LQNq · 2022-08-08
> > **Thank you for the reply**
> >
> > Thank you for answering my questions.
> > Though I think there is still some space for making the paper more clear, I raised the score from 4 to 5.

---

### Official Review · Reviewer_s32b · 2022-07-11

**Rating:** 6
**Confidence:** 5
**Soundness:** 2 fair
**Presentation:** 3 good
**Contribution:** 3 good

**Summary:**

This paper proposes new insights into the analysis of Learning using privileged information  (LuPI) learner with access to intermediate time series data and  generalizes it to nonlinear prediction tasks in latent dynamical systems. The authors extend theoretical guarantees to the case where the map connecting latent variables and observations is known up to a linear transform. In addition, the paper proposes algorithms based on random features, and representation learning for the case when the map is unknown.

**Questions:**

What is the benefit of the approach for prediction if it is not experimented for long term prediction?

How would the approach fare for prediction using different prediction horizons?

Beyond bias/variance analysis, what is the significance of this approach?

Which kind of non-linearity makes the proposed approach non-longer valid?

What is the algorithmic complexity for its implementation and use in practical settings?

**Limitations:**

It would be good if the authors could discuss the impact of their bias/variance analysis on societal applications?

**Strengths And Weaknesses:**

The paper is well written and theoretically sounds. The paper is clear easy to follow and bring to the fore an interesting contribution.

Particularly, the fact that the authors extend the LuPTS framework to nonlinear models and prediction tasks in latent dynamical systems is an important step.

Furthermore, the authors prove that learning with privileged information leads to lower risk when the nonlinear map connecting latent variables and observations is known up to a linear transform. This is also an important contribution.

The bias/variance analysis is also timely and important. Also, the experimentation with real data namely the prediction of traffic volume and Alzheimer progression is critical to validating the approach.

However, the fact that the study is limited to predictions for a fixed time horizon given present observations limit the extend to which the approach could be validated and applied.

---

> ### Author Response · Authors · 2022-07-31
> **Response to review by Reviewer s32b**
>
> We thank s32b for their insightful feedback!
>
> **Applications with a fixed time horizon**
>
> The fixed-horizon setting is actually very common in applications. Other than the ADNI task (Section 4) to predict *2-year* progression of Alzheimer’s disease [1], important examples from healthcare include predicting *30-day* mortality, e.g., following surgery [2], and predicting *30 or 180-day* readmission of patients [3]. Outside of healthcare, examples include predicting user churn within a *fixed* interval [4],  predicting *yearly* crop yields based on satellite imagery of farms [5] or predicting the *final cost* of a construction project at the end of planning [6]. Privileged time-series information is available in all of these examples (daily/hourly patient vitals; intermediate customer interactions, daily/monthly satellite imagery, construction events and changes).
>
> **Benefit of the approach for prediction beyond long term prediction**
>
> In experiments, we demonstrate that there is a consistent benefit even if there is only a single privileged time point, where data comprises $X_1, X_2$ and $Y$ (see e.g., Fig 4a). The meaning of “long term” is domain dependent, but we see gains across varying horizons T (see comment below). More generally, the same approach could be used when the index t = 1, …, T does not even refer to time, but to, for example, a spatial coordinate.
>
> **Prediction using different prediction horizons**
>
> In Appendix F, we give several examples of varying the horizon T (e.g., Figure 16a). We see that the benefit of privileged information remains for increasing T but note that the hardness of the prediction problem also changes with T ($R^2$ goes down for all methods in Figure 16a when T increases).
>
> If the reviewer means predicting outcomes for different horizons using the same model, success would depend on the nature of the dynamical system. For our Theorem, we have not assumed that the system is stationary, but this would be required to use the same model for arbitrary horizons, without other assumptions. This is an interesting question for further study and we will include a comment about it in the discussion.
>
> **Significance beyond bias/variance analysis**
>
> Our main goal is to reduce variance in learning—to increase sample efficiency—so as to make more accurate predictions. Empirically, we see that this reduction is beneficial for reducing the prediction risk in almost all cases. In the experiments with neural networks applied to images, we also see that the approach has benefits in terms of recovering the underlying latent state space Z, both quantitatively (Fig 7a) and qualitatively (Fig 7b).
>
> **Which kind of non-linearity makes the proposed approach non-longer valid?**
>
> For Theorem 1, we only require that the observation function is injective. For learning an unknown representation $\Phi$, we must place further restrictions to get the same guarantee. The details will depend on the choice of estimator and the data generating process. For example, using Random Fourier Features (RFF) assumes that $\Phi$ is continuous. Hence, this approach would not work well for discontinuous processes. In experiments, we see benefits even when we cannot guarantee that our assumptions hold.
> It is important to note that this same limitation applies to classical learning—we use the same hypothesis class in both paradigms and if the hypothesis class cannot approximate the true function well, we cannot expect good results.
>
> **Algorithmic complexity**
>
> For training the kernel and random feature estimators, the complexity will be (roughly) T times the complexity of the matching classical estimator; one fits T linear regressions and T random feature transformations. At test time, the complexities of classical and privileged estimators are equal. For the privileged neural network estimators, the complexity is equivalent to training a single recurrent neural network with T time points. We stress that algorithmic complexity is not a focus of this work.
>
> **References**
>
> [1] Beltran, J. F., Wahba, B. M., Hose, N., Shasha, D., Kline, R. P., & Alzheimer’s Disease Neuroimaging Initiative. (2020). PloS one, 15(7), e0235663.
>
> [2] Karhade, A. V., Thio, Q. C., Ogink, P. T., Shah, A. A., Bono, C. M., Oh, K. S., ... & Schwab, J. H. (2019). Neurosurgery, 85(1), E83-E91.
>
> [3] Mortazavi, B. J., Downing, N. S., Bucholz, E. M., Dharmarajan, K., Manhapra, A., Li, S. X., ... & Krumholz, H. M. (2016). Circulation: Cardiovascular Quality and Outcomes, 9(6), 629-640.
>
> [4] Huang, B., Kechadi, M. T., & Buckley, B. (2012). Expert Systems with Applications, 39(1), 1414-1425.
>
> [5] You, J., Li, X., Low, M., Lobell, D., & Ermon, S. (2017, February). Thirty-First AAAI conference on artificial intelligence.
>
> [6] Williams, Trefor P., and Jie Gong. Automation in Construction 43 (2014): 23-29.

---

### Official Review · Reviewer_gvbk · 2022-07-12

**Rating:** 6
**Confidence:** 3
**Soundness:** 3 good
**Presentation:** 3 good
**Contribution:** 3 good

**Summary:**

The paper deals with the data generated by the following model.

A vector $z_1$ comes from some distribution. Then it is subjected to a chain of linear transformations. The vector $z_2$ is obtained as a linear transformation of $z_1$ (plus some noise), $z_3$ is a linear transformation of $z_2$ etc. Finally we get to $z_T$. Then $z_T$ generates a label $y$ by a linear transformation of $a$ (finite dimensional) feature mapping.

The vectors $z_t$ are never visible directly. There are two modes of access to them. In the first, "classical" scenario we get to see $x_1 = \Psi (x_1)$ where $\Psi$ is a linear transformation. In the second, "privileged" scenario we get access to $x_2 = \Psi(z_2), x_3 = \Psi(z_3)$ etc.

We can have either classical or privileged information at the training stage, but only get classical information at the test stage. The paper compares two learning approaches.

In the first, classical approach we use Least Squares to regress y on a feature mapping of $x_1$. In the second, privileged approach we try reconstruct the whole model using least squares on the training set to find all transition linear transformations etc and get $y$. The paper shows that the variance of the later approach is lower than the variance of the former.

The advantage disappears if the feature mapping is infinite-dimensional and we get non-singular Gram matrices on the data.

The experimental results include reconstruction of the feature mapping using neural networks.

**Questions:**

None. The paper is clearly written.

**Limitations:**

Yes.

**Strengths And Weaknesses:**

I think this is an interesting result but its significance is somewhat limited. The authors very honestly show the key limitation for universal kernels. The result is certainly important for the dynamic systems community but I have doubts about wider ML audience. The very model appears limited (although it has important applications in neurophysiology).

Typos/suggestions

Page 3, line 93 on the form -> of the form ?

I also find the title somewhat misleading. This is not what is usually meant by a time series. "Dynamic systems" would be better.

---

> ### Author Response · Authors · 2022-07-31
> **Response to review by Reviewer gvbk**
>
> We thank gvbk for their insightful feedback!
>
> **An important correction**
>
> Regarding the summary by reviewer gvbk, we like to point out that $\Psi$ is *not* generally a linear transformation. Our work precisely deals with generalizing to the case where this transformation is nonlinear (see comment on l.86 in the original submission).
>
> **Interesting but limited significance; Limitation for universal kernels; Importance for wider ML audience; The very model appears limited**
>
> We appreciate the reviewer’s concern but argue that our setting is actually very general. Our method is applicable whenever predictions are made at a baseline time point about target outcomes at a fixed follow-up time and information is collected in between. Other than the ADNI task (Section 4) to predict 2-year progression of Alzheimer’s disease [1], important examples from healthcare include predicting 30-day mortality, e.g., following surgery [2], and predicting readmission of patients [3]. Outside of healthcare, examples include predicting user churn within a fixed interval [4], predicting yearly crop yields based on satellite imagery of farms [5] or predicting the final cost of a construction project at the end of planning [6]. Data that can be regarded as privileged time-series information is available in all of these examples (daily/hourly patient vitals; intermediate customer interactions, daily/monthly satellite imagery, construction events and changes). In addition we would like to highlight that the primary goal of our work is to make an accurate prediction of the outcome rather than learning the dynamical system.
>
> We mention universal kernels as they are a popular choice in combination with linear models, not because of their importance in connection to our method. In real-world applications random feature methods often outperform universal kernels. Consequently, Proposition 1 is not a great limitation in terms of predictive accuracy. As the alternatives (random features and neural networks) show great benefits in empirical results, we do not view this as a key limitation of our approach.
>
> Regarding limitations, the model we learn is a combination of a representation $\Phi$ and linear hypothesis. This is the most widely used model type in machine learning today. The only difference is that we make use of more than standard input-output pairs to learn it. In Theorem 1, we prove that doing so is more efficient when the data comes from a latent dynamical system as described in Assumption 1. It can be argued that, for the examples mentioned above (mortality, crop yield, construction cost), the causal mechanisms driving the system (e.g., disease progression) are not directly observed and therefore best represented as latent variables. Moreover, this does not rule out that our algorithm, or variants thereof, is more efficient than classical learning even when the assumptions are violated. We point out in l.262–265 of the original manuscript that we always observe a reduction in variance through the use of privileged time-series information. Indeed, in real-world experiments, we cannot guarantee that Assumption 1 holds, but nevertheless we witness consistent improvements using our method in all of them. This indicates that the model is not so limited after all.
>
> **Somewhat misleading title; Not what is usually meant by a time series; "Dynamic systems" would be better.**
>
> We thank the reviewer for this suggestion. In the title, “time-series” refers to the type of data used for learning, which is certainly a time series, not to a particular learning problem. In our work, we reframe classical prediction problems (where observations of inputs and the target label are separated in time) as problems of learning from time series data. The reviewer is right that, in Theorem 1, we assume that the time series is generated by a (latent) dynamical system. However, in experiments, we show that the method works also when the data generating process for the time series is unknown and may not match the assumptions of Theorem 1.
>
> **References**
>
> [1] Beltran, J. F., Wahba, B. M., Hose, N., Shasha, D., Kline, R. P., & Alzheimer’s Disease Neuroimaging Initiative. (2020). PloS one, 15(7), e0235663.
>
> [2] Karhade, A. V., Thio, Q. C., Ogink, P. T., Shah, A. A., Bono, C. M., Oh, K. S., ... & Schwab, J. H. (2019). Neurosurgery, 85(1), E83-E91.
>
> [3] Mortazavi, B. J., Downing, N. S., Bucholz, E. M., Dharmarajan, K., Manhapra, A., Li, S. X., ... & Krumholz, H. M. (2016). Circulation: Cardiovascular Quality and Outcomes, 9(6), 629-640.
>
> [4] Huang, B., Kechadi, M. T., & Buckley, B. (2012). Expert Systems with Applications, 39(1), 1414-1425.
>
> [5] You, J., Li, X., Low, M., Lobell, D., & Ermon, S. (2017, February). Thirty-First AAAI conference on artificial intelligence.
>
> [6] Williams, Trefor P., and Jie Gong. Automation in Construction 43 (2014): 23-29.

---

### Official Review · Reviewer_oWQK · 2022-07-12

**Rating:** 7
**Confidence:** 3
**Soundness:** 3 good
**Presentation:** 4 excellent
**Contribution:** 3 good

**Summary:**

Through both rigorous theoretical analysis and empirical findings, this paper extends results on Learning using Privileged Information to general time series governed by linear latent dynamics and potentially nonlinear observation models. Under this general class of possible models, the authors demonstrate through learning theoretic analysis that learning with privileged (data that is available at training time but not at inference time) time series always leads to lower or equivalent risk, given that the nonlinear map from observations to latent variables is known up to a linear mapping. They then extend their results to the unknown mappings via an argument based on the use of random feature maps. Finally, the authors propose to approximate unknown mappings with deep learning architectures. In their empirical study, they demonstrate that the predicted learners with privileged information generally outperform baselines in terms of sample efficiency.

**Questions:**

- Sample efficiency is qualitative and subjective. In Fig 4(b) OLS seems to be outperforming all baselines in terms of "sample-efficiency" in contradiction to the text. So what precisely is meant by sample efficiency here? Could that be made more rigorous?
- Figure 7(a) is not readable, and the conclusion is mostly not accessible. Could you please revise and explain?
- Will the experiments be made available and reproducible? There is no mention of this in the paper.

**Limitations:**

- I would recommend to spend much more time on the motivation, and amplify this in the experiments. For example, how does this setting connect to images (which is presented in the experiments)?
- I would recommend to make the findings around L151 more precise instead of deferring to the appendix as I believe this is one of the key findings in the paper.

Minor point:
- L66 typo in "respect"

**Strengths And Weaknesses:**

The paper benefits both from strong theoretical analysis and rigorous empirical set-up. The results presented are relevant, and significantly expand understanding of learning using privileged time series (LuPTS) information. To the best of my knowledge, the paper proposes both novel theoretical findings and explores a set of new approaches for LuPTS. Finally, the paper is well-written, with clear and concise use of language.

My only concern about the paper is motivation. It is not clear to me, except some cases alluded to in the introduction to the paper, in which applications one would be interested in using privileged information. Unfortunately, the empirical study also does not motivate a single application but rather transforms existing time series applications such that the problem definition fits. I believe the readers would benefit greatly from a discussion of motivation and potential impact.

---

> ### Author Response · Authors · 2022-07-31
> **Response to review by Reviewer oWQK**
>
> We thank oWQK for their insightful feedback!
>
> **Motivation and applicability**
>
> Our method is applicable when predictions are made at a baseline time point about target outcomes at a fixed follow-up time with data collected in between. The ADNI task (Sec 4) to predict 2-year progression of Alzheimer’s disease (AD), is an example of this [1]. Other examples from healthcare include predicting 30-day mortality, e.g., following surgery [2], and predicting readmission of patients [3]. Outside of healthcare, predicting user churn within a fixed interval [4], predicting yearly crop yields based on satellite imagery of farms [5] or predicting the final cost of a construction project at the end of planning [6]. Privileged time-series information is available in all of these examples (regular patient vitals; intermediate customer interactions, monthly satellite imagery, construction events). We believe it is a strength of our method that it is useful in many settings, as seen in our empirical results.
>
> **Defining sample efficiency; OLS in Fig 4(b)**
>
> Sample efficiency classically refers to the ability of a statistical estimator to achieve smaller error/risk than other estimators with the same number of samples. For learning algorithms, this typically refers to how the expected error of a learning algorithm varies with the number of training samples [7, Chapter 20] — a learner is more efficient if the expected risk $\bar{R}_D$ is smaller for the same number of samples $m = |D|$. Theorem 1 shows that privileged learning is more efficient than classical learning under appropriate conditions. We have clarified the connection between risk and sample efficiency on l.65–72 and l.125–128 in the revision. Empirically, we estimate efficiency by comparing the prediction error for different estimators as we vary the number of training samples. As the reviewer correctly points out, in Fig 4(b), OLS is more efficient than non-linear estimators *for very small sample sizes*, and the text (l.241–245 in the submission) acknowledges this, but it is less efficient at larger sample sizes due to large bias. Linear LuPTS has lower variance than OLS but higher bias here (see Fig 6).
>
> **Readability and conclusion of Fig 7(a)**
>
> We agree that the readability of Fig 7(a) can be improved. The revision includes an updated version of the figure with the shaded marks removed, such that only mean values are displayed. We provide a clear conclusion about the content of Figure 7(a) in l.283 (original submission), by stating that the privileged learners are more accurate in their predictions (x-Axis) and also produce higher SVCCA coefficients (y-axis) on average.
>
> **Code availability**
>
> Yes, the experiments will be made available in a code repository when the paper is made public; we have added a footnote on page 6.
>
> **I would recommend to spend much more time on the motivation; How does this setting connect to images?**
>
> In the camera-ready version, we will use the additional space to make our motivating applications more visible (see answer to first question). The problem of predicting crop yields from satellite imagery is an example of how our setting is connected to images [5]. Here, both baseline inputs and privileged information are satellite images taken of the same farm and the target outcome is yearly crop yield.
>
> **Make the findings around L151 more precise instead of deferring to the appendix**
>
> We thank Reviewer oWQK for recognizing the value of this result. Due to the page limit, we prioritized results concerning finite-sample properties (Theorem 1 and empirical results), since this is our main focus, over the asymptotic results discussed around l.151 in the original submission. Nevertheless, we believe that proving consistency of the random features method is an important step in justifying its use; we can trust the method to do the right thing as we increase $m$. We plan to expand the statements in the camera-ready version.
>
> **References**
>
> [1] Beltran, J. F., Wahba, B. M., Hose, N., Shasha, D., Kline, R. P., & Alzheimer’s Disease Neuroimaging Initiative. (2020). PloS one, 15(7), e0235663.
>
> [2] Karhade, A. V., Thio, Q. C., Ogink, P. T., Shah, A. A., Bono, C. M., Oh, K. S., ... & Schwab, J. H. (2019). Neurosurgery, 85(1), E83-E91.
>
> [3] Mortazavi, B. J., Downing, N. S., Bucholz, E. M., Dharmarajan, K., Manhapra, A., Li, S. X., ... & Krumholz, H. M. (2016). Circulation: Cardiovascular Quality and Outcomes, 9(6), 629-640.
>
> [4] Huang, B., Kechadi, M. T., & Buckley, B. (2012). Expert Systems with Applications, 39(1), 1414-1425.
>
> [5] You, J., Li, X., Low, M., Lobell, D., & Ermon, S. (2017, February). Thirty-First AAAI conference on artificial intelligence.
>
> [6] Williams, Trefor P., and Jie Gong. Automation in Construction 43 (2014): 23-29.
>
> [7] Dekking, F. M., Kraaikamp, C., Lopuhaä, H. P., & Meester, L. E. (2005). A Modern Introduction to Probability and Statistics: Understanding why and how (Vol. 488). London: Springer.

---

### Author Response · Authors · 2022-07-31
**Thanks to reviewers and general comments**

We thank the reviewers for their insightful feedback and for recognizing the merits of our theoretical (as pointed out by reviewer oWQK, gvbk, s32b) and empirical contributions (oWQK, s32b) and the potential value of our method to the community (all reviewers).

Reviewers asked about the motivation and generality of our work and on the notion of efficiency. We respond to each reviewer in turn but would like to make the following general clarifications:

* Our method is a tool for learning to make predictions of a single outcome at a fixed horizon. In addition to the applications mentioned in the paper, there are many more examples: predicting *30-day* mortality [1] or readmissions for patients [2]; predicting churn of users of an online service [3]; predicting *yearly* crop yields based on satellite imagery of farms [4]; predicting the *final* cost of a construction project at the end of planning [5]. Data that can be regarded as privileged time-series information is available for learning in all of these examples (daily/hourly patient vitals; intermediate user interactions, daily satellite imagery, construction events and changes), but not at the time of prediction.

- For problems like this, classification and regression models are typically trained only on features that are available also at test time. Karlsson et al. (2022) showed that when privileged information is available and is related linearly to baseline features and the outcome, learning also from that reduces the expected error (increases efficiency). In this work, we derive methods which make use of this also in *nonlinear* prediction tasks, greatly extending the generality of the idea. We plan to include the examples mentioned above in the introduction of the camera-ready version of the manuscript.

* Reviewers asked about the precise meaning of “efficiency”. Following tradition, see e.g., [6, chapter 20], we define an efficient estimator to be one that achieves the same error as another estimator using a smaller number of samples or, equivalently, a smaller error using the same number of samples. Our theoretical result proves that, in the right conditions, a learner using privileged information is at least as efficient (has at least as small a risk) as a comparable learner which does not use it—for any fixed training sample size. We have clarified the connection between risk (equation 2) and efficiency on l.65–72 and l.125–128 in the revision.

Authors

**References**

[1] Karhade, A. V., Thio, Q. C., Ogink, P. T., Shah, A. A., Bono, C. M., Oh, K. S., ... & Schwab, J. H. (2019). Neurosurgery, 85(1), E83-E91.

[2] Mortazavi, B. J., Downing, N. S., Bucholz, E. M., Dharmarajan, K., Manhapra, A., Li, S. X., ... & Krumholz, H. M. (2016). Circulation: Cardiovascular Quality and Outcomes, 9(6), 629-640.

[3] Huang, B., Kechadi, M. T., & Buckley, B. (2012). Expert Systems with Applications, 39(1), 1414-1425.

[4] You, J., Li, X., Low, M., Lobell, D., & Ermon, S. (2017, February). Thirty-First AAAI conference on artificial intelligence.

[5] Williams, Trefor P., and Jie Gong. Automation in Construction 43 (2014): 23-29.

[6] Dekking, F. M., Kraaikamp, C., Lopuhaä, H. P., & Meester, L. E. (2005). A Modern Introduction to Probability and Statistics: Understanding why and how (Vol. 488). London: Springer.

---

### Author Response · Authors · 2022-08-05
**Discussion**

Dear reviewers and chairs,

Please let us know if there are any further clarifications needed concerning our paper after the rebuttal.
We would be happy to answer your questions.

Authors

---

### Meta-Review · Area_Chair_qxiY · 2022-08-26

**Recommendation:** Accept
**Confidence:** Certain

**Metareview:**

This paper considers a particular setting of time series prediction with privileged information. A special case can be described as predicting x(t+k) from x(t). At training time one is also given x(t+1), x(t+2), ..., x(t+k-1) and a latent dynamics is assumed. The paper presents a learning algorithm that leverages privileged info at train time, provides rigorous theoretical analysis of this algorithm and  convincing numerical experiments. This paper is definitely of interest to ML community and would serve as an interesting contribution to the conference.

**Award:**

No

---

### Decision · Program_Chairs · 2022-09-14

Accept